# Learning One-hidden-layer Neural Networks with Landscape Design

**Rong Ge**
Computer Science Department
Duke University
`rongge@cs.duke.edu`

**Jason D. Lee**
Data Sciences and Operations Department,
University of Southern California
`jasonlee@marshall.usc.edu`

**Tengyu Ma**
Facebook AI Research
`tengyuma@cs.stanford.edu`

## Abstract

We consider the problem of learning a one-hidden-layer neural network: we assume the input $x \in \mathbb{R}^d$ is from Gaussian distribution and the label $y = a^\top \sigma(Bx) + \xi$, where $a$ is a nonnegative vector in $\mathbb{R}^m$ with $m \le d$, $B \in \mathbb{R}^{m \times d}$ is a full-rank weight matrix, and $\xi$ is a noise vector. We first give an analytic formula for the population risk of the standard squared loss and demonstrate that it implicitly attempts to decompose a sequence of low-rank tensors simultaneously.

Inspired by the formula, we design a non-convex objective function $G(\cdot)$ whose landscape is guaranteed to have the following properties:

1. All local minima of $G$ are also global minima.
2. All global minima of $G$ correspond to the ground truth parameters.
3. The value and gradient of $G$ can be estimated using samples.

With these properties, stochastic gradient descent on $G$ provably converges to the global minimum and learn the ground-truth parameters. We also prove finite sample complexity results and validate the results by simulations.

## 1 Introduction

Scalable optimization has played an important role in the success of deep learning, which has immense applications in artificial intelligence. Remarkably, optimization issues are often addressed through designing new models that make the resulting training objective functions easier to be optimized. For example, over-parameterization (Livni et al., 2014), batch-normalization (Ioffe & Szegedy, 2015), and residual networks (He et al., 2016a;b) are often considered as ways to improve the optimization landscape of the resulting objective functions.

How do we design models and objective functions that allow efficient optimization with guarantees? Towards understanding this question in a principled way, this paper studies learning neural networks with one hidden layer. Roughly speaking, we will show that when the input is from Gaussian distribution and under certain simplifying assumptions on the weights, we can design an objective function $G(\cdot)$, such that

[a] all local minima of $G(\cdot)$ are global minima

[b] all the global minima are the desired solutions, namely, the ground-truth parameters (up to permutation and some fixed transformation).

We note that designing such objective functions is challenging because 1) the natural $\ell_2$ loss objective does have bad local minimum, and 2) due to the permutation invariance[1], the objective function inherently has to contain an exponential number of isolated local minima.

---

[1] Permuting the rows of $B^\star$ and the coordinates of $a^\star$ correspondingly preserves the functionality of the network.

## 1.1 SETUP AND KNOWN ISSUES WITH PROPER LEARNING

We aim to learn a neural network with a one-hidden-layer using a non-convex objective function. We assume input $x$ comes from Gaussian distribution and the label $y$ comes from the model

$$y = a^{\star\top}\sigma(B^\star x) + \xi \tag{1.1}$$

where $a^\star \in \mathbb{R}^m, B^\star \sim \mathbb{R}^{m\times d}$ are the ground-truth parameters, $\sigma(\cdot)$ is a element-wise non-linear function, and $\xi$ is a noise vector with zero mean. Here we can without loss of generality assume $x$ comes from spherical Gaussian distribution $\mathcal{N}(0, \mathrm{Id}_{d\times d})$. [2]

For technical reasons, we will further assume $m \leq d$ and that $a^\star$ has non-negative entries.

The most natural learning objective is perhaps the $\ell_2$ loss function, given the additive noise. Concretely, we can parameterize with training parameters $a \in \mathbb{R}^m, B \sim \mathbb{R}^{m\times d}$ of the same dimension as $a^\star$ and $B^\star$ correspondingly,

$$\hat{y} = a^\top \sigma(Bx), \tag{1.2}$$

and then use stochastic gradient descent to optimize the $\ell_2$ loss function. In many parts of our paper, we consider $\sigma$ to be the ReLU function $\sigma(x) = \max\{x, 0\}$. In such settings we assume rows of $B^\star$ have norm 1 because $a^\star$ and rows of $B^\star$ can be scaled simultaneously without changing the model or the objective.

When we have enough training examples, we are effectively minimizing the following population risk with stochastic updates,

$$f(a, B) = \mathbb{E}\left[\|\hat{y} - y\|^2\right]. \tag{1.3}$$

However, **empirically** stochastic gradient descent **cannot** converge to the ground-truth parameters in the synthetic setting above when $\sigma(x) = \mathrm{ReLU}(x) = \max\{x, 0\}$, even if we have access to an infinite number of samples, and $B^\star$ is a orthogonal matrix. We also show this phenomena generalizes to the case when $\sigma(x)$ is the sigmoid function and the learned network also have the same architecture. Such empirical results have been reported in Livni et al. (2014) previously, and we also provide our version in Figure 1 and Figure 2 of Section 4. This is consistent with observations and theory that over-parameterization is important for training neural networks successfully (Livni et al., 2014; Hardt et al., 2016; Soudry & Carmon, 2016).

These empirical findings suggest that the population risk $f(a, B)$ has spurious local minima with inferior error compared to that of the global minimum. This phenomenon occurs even if we assume we know $a^\star$ or $a^\star = \mathbf{1}$ is merely just the all one's vector. Empirically, such landscape issues seem to be alleviated by over-parameterization. By contrast, our method described in the next section does not require over-parameterization and might be suitable for applications that demand the recovery of the true parameters.

## 1.2 OUR CONTRIBUTIONS

Towards learning with the same number of training parameters as the ground-truth model, we first study the landscape of the population risk $f(\cdot)$ and give an analytic formula for it — as an explicit function of the ground-truth parameters and training parameters with the randomness of the data being marginalized out. The formula in equation (2.3) shows that $f(\cdot)$ is implicitly attempting to solve simultaneously an infinite number of low-rank tensor decomposition problems with commonly shared components.

Inspired by the formula, we design a new training model whose associated loss function — named $f'$ and formally defined in equation (2.5) — corresponds to the loss function for decomposing a matrix (2-nd order tensor) and a 4-th order tensor (Theorem 2.2). Empirically, stochastic gradient descent on $f'$ learns the network as shown Section 4.

Despite the empirical success of $f'$, we still lack a provable guarantee on the landscape of $f'$. The second contribution of the paper is to design a more sophisticated objective $G(\cdot)$ whose landscape

---

[2]This is because if $x \sim N(0, \Sigma)$, then we can whiten the data by taking $x' = \Sigma^{-1/2}x$ and define $B^{\star\prime} = B\Sigma^{1/2}$. We note that $B^{\star\prime}x' = Bx$ and therefore we maintain the functionality of the model.

is provably nice — all the local minima of $G(\cdot)$ are proven to be global, and they correspond to the permutation of the true parameters. See Theorem 2.3.

Moreover, the value and the gradient of $G$ can be estimated using samples, and there are no constraints in the optimization. These allow us to use straightforward SGD (see guarantees in Ge et al. (2015); Jin et al. (2017)) to optimize $G(\cdot)$ and converge to a local minimum, which is also a global minimum (Corollary 2.4).

Finally, we also prove a finite-sample complexity result. We will show that with a polynomial number of samples, the empirical version of $G$ share almost the same landscape properties as $G$ itself (Theorem 2.7). Therefore, we can also use an empirical version of $G$ as a surrogate in the optimization.

## 1.3 RELATED WORK

The work of Arora et al. (2014) is one of the early results on provable algorithms for learning deep neural networks, where the authors give an algorithm for learning deep generative models with sparse weights. Livni et al. (2014), Zhang et al. (2016; 2017b), and Daniely et al. (2016) study the learnability of special cases of neural networks using ideas from kernel methods. Janzamin et al. (2015) give a polynomial-time algorithm for learning one-hidden-layer neural networks with twice-differentiable activation function and known input distributions. Their approach uses the idea of score function to estimate the high order tensors related to the true components, and then apply tensor decompositions to recover the true parameters. When applied to Gaussian input distribution, the score function becomes Hermite polynomials.

A series of recent papers study the theoretical properties of non-convex optimization algorithms for one-hidden-layer neural networks. Brutzkus & Globerson (2017) and Tian (2017) analyze the landscape of the population risk for one-hidden-layer neural networks with Gaussian inputs under the assumption that the weights vector associated to each hidden variable (that is, the filters) have disjoint supports. Li & Yuan (2017) proves that stochastic gradient descent recovers the ground-truth parameters when the parameters are known to be close to the identity matrix. Zhang et al. (2017a) study the optimization landscape of learning one-hidden-layer neural networks with a specific activation function, and they design a specific objective function that can recover a single column of the weight matrix. Zhong et al. (2017) study the convergence of non-convex optimization from a good initializer that is produced by tensor methods. Our algorithm works for a large family of activation functions (including ReLU) and any full-rank weight matrix. To our best knowledge, we give the first global convergence result for gradient-based methods for our general setting. [3]

The optimization landscape properties have also been investigated on simplified neural networks models. Kawaguchi (2016) shows that the landscape of deep neural nets does not have bad local minima but has degenerate saddle points. Hardt & Ma (2017) show that re-parametrization using identity connection as in residual networks He et al. (2016a) can remove the degenerate saddle points in the optimization landscape of deep linear residual networks. Soudry & Carmon (2016) show that an over-parameterized neural network does not have bad differentiable local minimum. Hardt et al. (2016) analyze the power of over-parameterization in a linear recurrent network (which is equivalent to a linear dynamical system.)

The optimization landscape has also been analyzed for other machine learning problems, including SVD/PCA phase retrieval/synchronization, orthogonal tensor decomposition, dictionary learning, matrix completion, matrix sensing Baldi & Hornik (1989); Srebro & Jaakkola (2013); Ge et al. (2015); Sun et al. (2015); Bandeira et al. (2016); Ge et al. (2016); Bhojanapalli et al. (2016); Ge et al. (2017). Our analysis techniques build upon that for tensor decomposition in Ge et al. (2015) — we add two additional regularization terms to deal with spurious local minimum caused by the weights $a^\star$ and to remove the constraints.

**Notations:** We use $\|\cdot\|$ to denote the Euclidean norm of a vector and spectral norm of a matrix. We use $\|\cdot\|_F$ to denote the Frobenius/Euclidean norm of a matrix or high-order tensor. For a vector $x$, let $\|x\|_0$ denotes its infinity norm and for a matrix $A$, let $|A|_0$ be a shorthand for $\|\text{vec}(A)\|_0$ where $\text{vec}(A)$ is the vectorization of $A$.

---

[3]The work of Janzamin et al. (2015); Zhong et al. (2017) are closely related, but they require tensor decomposition as the algorithm/initialization.

We use $A \otimes B$ to denote the Kronecker product of $A$ and $B$, and $A^{\otimes k}$ is a shorthand for $A \otimes \cdots \otimes A$ where $A$ appears $k$ times. For vectors $a \otimes b$ and $a^{\otimes k}$ denote the tensor product. We denote the identity matrix in dimension $d \times d$ by $\mathrm{Id}_{d \times d}$, or $\mathrm{Id}$ when the dimension is clear from the context. We will define other notations when we first use them.

## 2 MAIN RESULTS

### 2.1 CONNECTING $\ell_2$ POPULATION RISK WITH TENSOR DECOMPOSITION

We first show that a natural $\ell_2$ loss for the one-hidden-layer neural network can be interpreted as simultaneously decomposing tensors of different orders.

A straightforward approach of learning the model (1.1) is to parameterize the prediction by

$$\hat{y} = a^\top \sigma(Bx), \tag{2.1}$$

where $a \in \mathbb{R}^d, B \sim \mathbb{R}^{m \times d}$ are the training parameters. Naturally, we can use $\ell_2$ as the empirical loss, which means the population risk is

$$f(a, B) = \mathbb{E}\left[\|\hat{y} - y\|^2\right]. \tag{2.2}$$

Throughout the paper, we use $b_1^{\star\top}, \ldots, b_m^{\star\top}$ to denote the row vectors of $B^\star$ and similarly for $B$.

That is, we have $B = \begin{bmatrix} b_1^\top \\ \vdots \\ b_m^\top \end{bmatrix}$ and $B^\star = \begin{bmatrix} b_1^{\star\top} \\ \vdots \\ b_m^{\star\top} \end{bmatrix}$. Let $a_i$ and $a_i^\star$'s be the coordinates of $a$ and $a^\star$ respectively.

We give the following analytic formula for the population risk defined above.

**Theorem 2.1.** *Assume vectors $b_i, b_i^\star$'s are unit vectors. Then, the population risk $f$ defined in equation* (2.2) *satisfies that*

$$f(a, B) = \sum_{k \in \mathbb{N}} \hat{\sigma}_k^2 \left\| \sum_{i \in [m]} a_i^\star b_i^{\star \otimes k} - \sum_{i \in [m]} a_i b_i^{\otimes k} \right\|_F^2 + \mathrm{const}. \tag{2.3}$$

*where $\hat{\sigma}_k$ is the $k$-th Hermite coefficient of the function $\sigma$. See section A.1 for a short introduction of Hermite polynomial basis.* [4]

*Connection to tensor decomposition:* We see from equation (2.3) that the population risk of $f$ is essentially an average of infinite number of loss functions for tensor decomposition. For a fixed $k \in \mathbb{N}$, we have that the $k$-th summand in equation (2.3) is equal to (up to the scaling factor $\hat{\sigma}_k^2$)

$$f_k \triangleq \|T_k - \sum_{i \in [m]} a_i b_i^{\otimes k}\|_F^2. \tag{2.4}$$

where $T_k = \sum_{i \in [m]} a_i^\star b_i^{\star \otimes k}$ is a $k$-th order tensor in $(\mathbb{R}^d)^{\otimes k}$. We note that the objective $f_k$ naturally attempts to decompose the $k$-order rank-$m$ tensor $T_k$ into $m$ rank-1 components $a_1 b_i^{\otimes k}, \ldots, a_m b_m^{\otimes k}$.

The proof of Theorem 2.1 follows from using techniques in Hermite Fourier analysis, which is deferred to Section A.2.

**Issues with optimizing $f$:.** It turns out that optimizing the population risk using stochastic gradient descent is empirically difficult. Figure 1 shows that in a synthetic setting where the noise is zero, the test error empirically doesn't converge to zero for sufficiently long time with various learning rate schemes, even if we are using fresh samples in iteration. This suggests that the landscape of the population risk has some spurious local minimum that is not a global minimum. See Section 4 for more details on the experiment setup.

---

[4] When $\sigma = ReLU$, we have that $\hat{\sigma}_0 = \frac{1}{\sqrt{2\pi}}, \hat{\sigma}_1 = \frac{1}{2}$. For $n \geq 2$ and even, $\hat{\sigma}_n = \frac{((n-3)!!)^2}{\sqrt{2\pi n!}}$. For $n \geq 2$ and odd, $\hat{\sigma}_n = 0$.

**An empirical fix:.** Inspired by the connection to tensor decomposition objective described earlier in the subsection, we can design a new objective function that takes exactly the same form as the tensor decomposition objective function $f_2 + f_4$. Concretely, let's define $\hat{y}' = a^\top \gamma(Bx)$ where $\gamma = \hat{\sigma_2}h_2 + \hat{\sigma_4}h_4$ and $h_2(t) = \frac{1}{\sqrt{2}}(t^2 - 1)$ and $h_4(t) = \frac{1}{\sqrt{24}}(t^4 - 6t^2 + 3)$ are the 2nd and 4th normalized probabilists' Hermite polynomials Wikipedia (2017a). We abuse the notation slightly by using the same notation to denote the its element-wise application on a vector. Now for each example we use $\|\hat{y}' - y\|^2$ as loss function. The corresponding population risk is

$$f'(a, B) = \mathbb{E}\left[\|\hat{y}' - y\|^2\right] . \tag{2.5}$$

Now by an extension of Theorem 2.1, we have that the new population risk is equal to the $\hat{\sigma}_2^2 f_2 + \hat{\sigma}_4^2 f_4$.

**Theorem 2.2.** *Let $f'$ be defined as in equation (2.5) and $f_2$ and $f_4$ be defined in equation (2.4). Assume $b_i, b_i^\star$'s are unit vectors. Then, we have*

$$f' = \hat{\sigma}_2^2 f_2 + \hat{\sigma}_4^2 f_4 + \text{const} \tag{2.6}$$

It turns out stochastic gradient descent on the objective $f'(a, B)$ (with projection to the set of matrices $B$ with row norm 1) converges empirically to the ground truth $(a^\star, B^\star)$ or one of its equivalent permutations. (See Figure 3.) However, we don't know of any existing work for analyzing the landscape of the objective $f'$ (or $f_k$ for any $k \geq 3$). We conjecture that the landscape of $f'$ doesn't have any spurious local minimum under certain mild assumptions on $(a^\star, B^\star)$. Despite recent attempts on other loss functions for tensor decomposition Ge & Ma (2017), we believe that analyzing $f'$ is technically challenging and its resolution will be potentially enlightening for the understanding landscape of loss function with permutation invariance. See Section 4 for more experimental results.

## 2.2 Landscape design for orthogonal $B^\star$

The population risk defined in equation (2.5) — though works empirically for randomly generated ground-truth $(a^\star, B^\star)$ — doesn't have any theoretical guarantees. It's also possible that when $(a^\star, B^\star)$ are chosen adversarially or from a different distribution, SGD no longer converges to the true parameters.

To solve this problem, we design another objective function $G(\cdot)$, such that the optimizer of $G(\cdot)$ still corresponds to the ground-truth, and $G()$ has provably nice landscape — all local minima of $G()$ are global minima.

In this subsection, for simplicity, we work with the case when $B^\star$ is an orthogonal matrix and state our main result. The discussion of the general case is deferred to the end of this Section and Section C.

We define our objective function $G(B)$ as

$$G(B) \triangleq \text{sign}(\hat{\sigma}_4) \mathbb{E}\left[y \cdot \sum_{j,k \in [d], j \neq k} \phi(b_j, b_k, x)\right] - \mu \, \text{sign}(\hat{\sigma}_4) \mathbb{E}\left[y \cdot \sum_{j \in [d]} \varphi(b_j, x)\right]$$

$$+ \lambda \sum_{i=1}^{m} (\|b_i\|^2 - 1)^2 \tag{2.7}$$

where $\varphi(\cdot, \cdot)$ is defined as

$$\varphi(v, x) = \frac{1}{8}\|v\|^4 - \frac{1}{4}(v^\top x)^2 \|v\|^2 + \frac{1}{24}(v^\top x)^4 . \tag{2.8}$$

and $\phi(\cdot, \cdot, \cdot)$ is defined as

$$\phi(v, w, x) = \frac{1}{2}\|v\|^2 \|w\|^2 + \langle v, w \rangle^2 - \frac{1}{2}\|w\|^2 (v^\top x)^2 - \frac{1}{2}\|v\|^2 (w^\top x)^2$$

$$+ 2(v^\top x)(w^\top x)v^\top w + \frac{1}{2}(v^\top x)^2 (w^\top x)^2 . \tag{2.9}$$

 The rationale behind of the choices of $\phi$ and $\varphi$ will only be clearer and relevant in later sections. For now, the only relevant property of them is that both are smooth functions whose derivatives are easily computable.

We remark that we can sample $G(\cdot)$ using the samples straightforwardly — it's defined as an average of functions of examples and the parameters. We also note that only parameter $B$ appears in the loss function. We will infer the value of $a^\star$ using straightforward linear regression after we get the (approximately) accurate value of $B^\star$.

Due to technical reasons, our method only works for the case when $a_i^\star > 0$ for every $i$. We will assume this throughout the rest of the paper. The general case is left for future work. Let $a_{\max}^\star = \max a_i^\star$, $a_{\min}^\star = \min a_i^\star$, and $\kappa^\star = \max a_i^\star / \min a_i^\star$. Our result will depend on the value of $\kappa^\star$. Essentially we treat $\kappa^\star$ as an absolute constant that doesn't scale in dimension. The following theorem characterizes the properties of the landscape of $G(\cdot)$.

**Theorem 2.3.** *Let $c$ be a sufficiently small universal constant (e.g. $c = 0.01$ suffices) and suppose the activation function $\sigma$ satisfies $\hat{\sigma}_4 \neq 0$. Assume $\mu \leq c/\kappa^\star$, $\lambda \geq c^{-1} a_{\max}^\star$, and $B^\star$ is an orthogonal matrix. The function $G(\cdot)$ defined as in equation (2.7) satisfies that*

1. *A matrix $B$ is a local minimum of $G$ if and only if $B$ can be written as $B = DPB^\star$ where $P$ is a permutation matrix and $D$ is a diagonal matrix with $D_{ii} \in \{\pm 1 \pm O(\mu a_{\max}^\star / \lambda)\}$.[5] Furthermore, this means that all local minima of $G$ are also global.*

2. *Any saddle point $B$ has a strictly negative curvature in the sense that $\lambda_{\min}(\nabla^2 G(B)) \geq -\tau_0$ where $\tau_0 = c \min\{\mu a_{\min}^\star / (\kappa^\star d), \lambda\}$*

3. *Suppose $B$ is an approximate local minimum in the sense that $B$ satisfies*

$$\|\nabla G(B)\| \leq \varepsilon \text{ and } \lambda_{\min}(\nabla^2 G(B)) \geq -\tau_0$$

   *Then $B$ can be written as $B = PDB^\star + EB^\star$ where $P$ is a permutation matrix, $D$ is a diagonal matrix satisfying the same bound as in bullet 1, and $|E|_\infty \leq O(\varepsilon/(\hat{\sigma}_4 a_{\min}^\star))$.*

   *As a direct consequence, $B$ is $O_d(\varepsilon)$-close to a global minimum in Euclidean distance, where $O_d(\cdot)$ hides polynomial dependency on $d$ and other parameters.*

The theorem above implies that we can learn $B^\star$ (up to permutation of rows and sign-flip) if we take $\lambda$ to be sufficiently large and optimize $G(\cdot)$ using stochastic gradient descent. In this case, the diagonal matrix $D$ in bullet 1 is sufficiently close to identity (up to sign flip) and therefore a local minimum $B$ is close to $B^\star$ up to permutation of rows and sign flip. The sign of each $b_i^\star$ can be recovered easily after we recover $a$ (see Lemma 2.5 below.)

SGD converges to a local minimum Ge et al. (2015) (under the additional property as established in bullet 2 above), which is also a global minimum for the function $G(\cdot)$. We will prove the theorem in Section B as a direct corollary of Theorem B.1. The technical bullet 2 and 3 of the theorem is to ensure that we can use SGD to converge to a local minimum as stated below.[6]

**Corollary 2.4.** *In the setting of Theorem 2.3, we can use stochastic gradient descent to optimize function $G(\cdot)$ (with fresh samples at each iteration) and converge to an approximate global minimum $B$ that is $\varepsilon$-close to a global minimum in time $\mathrm{poly}(d, 1/\varepsilon)$.*

After approximately recovering the matrix $B^\star$, we can also recover the coefficient $a^\star$ easily. Note that fixing $B$, we can fit $a$ using simply linear regression. For the ease of analysis, we analyze a slightly different algorithm. The lemma below is proved in Section D.

**Lemma 2.5.** *Given a matrix $B$ whose rows have unit norm, and are $\delta$-close to $B^\star$ in Euclidean distance up to permutation and sign flip with $\delta \leq 1/(2\kappa^\star)$. Then, we can give estimates $a, B'$ (using e.g., Algorithm 1) such that there exists a permutation $P$ where $\|a - Pa^\star\|_\infty \leq \delta a_{\max}^\star$ and $B'$ is row-wise $\delta$-close to $PB^\star$.*

The key step towards analyzing objective $G(B)$ is the following theorem that gives an analytic formula for $G(\cdot)$.

---

[5]More precisely, $|D_{ii}| = \sqrt{\frac{1}{1-\mu|\hat{\sigma}_4|a_i^\star/(\sqrt{6}\lambda)}}$

[6]In the most general setting, converging to a local minimum of a non-convex function is NP-hard.

**Theorem 2.6.** *The function $G(\cdot)$ satisfies*

$$G(B) = 2\sqrt{6}|\hat{\sigma}_4| \cdot \sum_{i \in [d]} a_i^\star \sum_{j,k \in [d], j \neq k} \langle b_i^\star, b_j \rangle^2 \langle b_i^\star, b_k \rangle^2 - \frac{|\hat{\sigma}_4|\mu}{\sqrt{6}} \sum_{i,j \in [d]} a_i^\star \langle b_i^\star, b_j \rangle^4 + \lambda \sum_{i=1}^m (\|b_i\|^2 - 1)^2$$

$$(2.10)$$

Theorem 2.6 is proved in Section A. We will motivate our design choices with a brief overview in Section 3 and formally analyze the landscape of $G$ in Section B (see Theorem B.1).

**Finite sample complexity bounds.** Extending Theorem 2.3, we can characterize the landscape of the empirical risk $\widehat{G}$, which implies that stochastic gradient on $\widehat{G}$ also converges approximately to the ground-truth parameters with polynomial number of samples.

**Theorem 2.7.** *In the setting of Theorem 2.3, suppose we use $N$ empirical samples to approximate $G$ and obtain empirical risk $\widehat{G}$. There exists a fixed polynomial $poly(d, 1/\varepsilon)$ such that if $N \geq poly(d, 1/\varepsilon)$, then with high probability the landscape of $\widehat{G}$ has the properties to that of $G$ in bullet 2 and 3 of Theorem 2.3.*

All of the results above assume that $B^\star$ is orthogonal. Since the local minimum are preserved by linear transformation of the input space, these results can be extended to the general case when $B^\star$ is not orthogonal but full rank (with some additional technicality) or the case when the dimension is larger than the number of neurons ($m < d$). See Section C.

## 3 OVERVIEW: LANDSCAPE DESIGN AND ANALYSIS

In this section, we present a general overview of ideas behind the design of objective function $G(\cdot)$. Inspired by the formula (2.3), in Section 3.1, we envision a family of possible objective functions for which we have unbiased estimators via samples. In Section 3.2, we pick a specific function that feeds our needs: a) it has no spurious local minimum; b) the global minimum corresponds to the ground-truth parameters.

### 3.1 WHICH OBJECTIVE CAN BE ESTIMATED BY SAMPLES?

Recall that in equation (2.2) of Theorem 2.1 we give an analytic formula for the straightforward population risk $f$. Although the population risk $f$ doesn't perform well empirically, the lesson that we learn from it help us design better objective functions. One of the key fact that leads to the proof of Theorem 2.1 is that for any continuous and bounded function $\gamma$, we have that

$$\mathbb{E}\left[y \cdot \gamma(b_i^\top x)\right] = \sum_{k \in \mathbb{N}} \hat{\gamma}_k \hat{\sigma}_k \left(\sum_{j \in [d]} a_j^\star \langle b_j^\star, b_i \rangle^k\right).$$

Here $\hat{\sigma}_k$ and $\hat{\gamma}_k$ are the $k$-th Hermite coefficient of the function $\sigma$ and $\gamma$. That is, letting $h_k$ the $k$-th normalized probabilists' Hermite polynomials Wikipedia (2017a) and $\langle \cdot, \cdot \rangle$ be the standard inner product between functions, we have $\hat{\sigma}_k = \langle h_k, \sigma \rangle$.

Note that $\gamma$ can be chosen arbitrarily to extract different terms. For example, by choosing $\gamma = h_k$, we obtain that

$$\mathbb{E}\left[y \cdot h_k(b_i^\top x)\right] = \hat{\sigma}_k \sum_{j \in [d]} a_j^\star \langle b_j^\star, b_i \rangle^k.$$

$$(3.1)$$

That is, we can always access functions forms that involves weighted sum of the powers of $\langle b_i^\star, b_i \rangle$, as in RHS of equation (3.1). Using a bit more technical tools in Fourier analysis (see details in Section A), we claim that most of the symmetric polynomials over variables $\langle b_i^\star, b_j \rangle$ can be estimated by samples:

*Claim 3.1 (informal).* For any polynomial $p()$ over a single variable, there exits a corresponding function $\phi^p$ such that

$$\mathbb{E}\left[y \cdot \phi^p(B, x)\right] = \sum_j a_j^\star \sum_i p(\langle b_j^\star, b_i \rangle)$$

$$(3.2)$$

Moreover, for an any polynomial $q(\cdot, \cdot)$ over two variables, there exists corresponding $\phi^q$ such that

$$\mathbb{E}\left[y \cdot \phi^q(B, x)\right] = \sum_j a_j^\star \sum_{i,k} q(\langle b_j^\star, b_i \rangle, \langle b_k^\star, b_i \rangle) \tag{3.3}$$

We will not prove these two general claims. Instead, we only focus on the formulas in Theorem A.5 and Theorem A.6, which are two special cases of the claims above.

Motivated by Claim A.3, in the next subsection, we will pick an objective function which has no spurious local minimum among those functional forms on the right-hand sides of equation (3.2) and (3.3).

## 3.2 WHICH OBJECTIVE HAS NO SPURIOUS LOCAL MINIMA?

As discussed briefly in the introduction, one of the technical difficulties to design and analyze objective functions for neural networks comes from the permutation invariance — if a matrix $B$ is a good solution, then any permutation of the rows of $B$ still gives an equally good solution (if we also permute the coefficients in $a$ accordingly). We only know of a very limited number of objective functions that guarantee to enjoy permutation invariance and have no spurious local minima Ge et al. (2015). We start by considering the objective function used in Ge et al. (2015),

$$\min\ P(B) = \sum_i \sum_{j \neq k} \langle b_i^\star, b_j \rangle^2 \langle b_i^\star, b_k \rangle^2$$
$$s.t.\ \forall i \in [d], \|b_i\| = 1 \tag{3.4}$$

Note that here we overload the notation by using $b_i^\star$'s to denote a set of fixed vectors that we wanted to recover and using $b_i$'s to denote the variables. Careful readers may notice that $P(B)$ doesn't fall into the family of functions that we described in the previous section (that is, RHS equation of (3.2) and (3.3)), because it lacks the weighting $a_i^\star$'s. We will fix this issue later in the subsection. Before that we first summarize the nice properties of the landscape of $P(B)$.

For the simplicity of the discussion, let's assume $B^\star$ forms an orthonormal matrix in the rest of the subsection. Then, any permutation and sign-flip of the rows of $B^\star$ leads to a global minimum of $P(\cdot)$ — when $B = SQB^\star$ with a permutation matrix $Q$ and a sign matrix $S$ (diagonal with $\pm 1$), we have that $P(B) = 0$ because one of $\langle b_i^\star, b_j \rangle^2$ and $\langle b_i^\star, b_k \rangle^2$ has to be zero for all $i, j, k$[7].

It turns out that these permutations/sign-flips of $B^\star$ are also the only local minima[8] of function $P(\cdot)$. To see this, notice that $P(B)$ is a degree-4 polynomial of $B$. Thus if we pick an index $s$ and fix every row except for $b_s$, then $P(B)$ is a quadratic function over unit vector $b_s$ – reduces to a smallest eigenvector problem. Eigenvector problems are known to have no spurious local minimum. Thus the corresponding function (w.r.t $b_s$) has no spurious local minimum. It turns out the same property still holds when we treat all the rows as variables and add the row-wise norm constraints.

However, there are two issues with using objective function $P(B)$. The obvious one is that it doesn't involve the coefficients $a_i^\star$'s and thus doesn't fall into the forms of equation (3.3). Optimistically, we would hope that for nonnegative $a_i^\star$'s the weighted version of $P$ below would also enjoy the similar landscape property

$$P'(B) = \sum_i a_i^\star \sum_{j \neq k} \langle b_i^\star, b_j \rangle^2 \langle b_i^\star, b_k \rangle^2$$

When $a_i^\star$'s are positive, indeed the global minimum of $P'$ are still just all the permutations of the $B^\star$.[9] However, when $\max a_i^\star > 2 \min a_i^\star$, we found that $P'$ starts to have spurious local minima . It seems that spurious local minimum often occurs when a row of $B$ is a linear combination of a smaller number of rows of $B^\star$. See Section F for a concrete example.

---

[7]Note that $B^\star$ is orthogonal, and $j \neq k$

[8]We note that since there are constraints here, we consider the local minimum on the manifold defined by the constraints.

[9]This is the main reason why we require $a^\star \geq 0$.

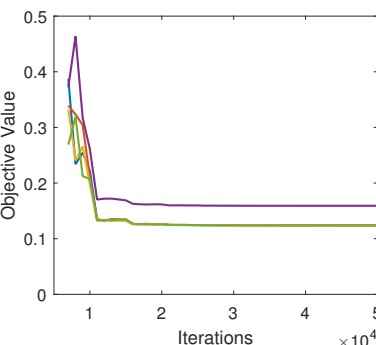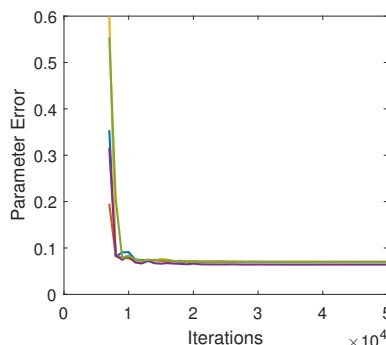

Figure 1: Data are generated by a network with ReLU activation without noise. The training model uses the same architecture. Left: the estimated population risk doesn't converge to zero. Right: the parameter error using the surrogate in equation (4.1).

To remove such spurious local minima, we add a regularization term below that pushes each row of $B$ to be close to one of the rows of $B^\star$,

$$R(B) = -\mu \sum_i a_i^\star \sum_j \langle b_i^\star, b_j \rangle^4 \tag{3.5}$$

We see that for each fixed $j$, the part in $R(B)$ that involves $b_j$ has the form $-\mu \sum_i a_i^\star \langle b_i^\star, b_j \rangle^4 = -\mu \langle \sum_i a_i^\star b_i^{\star \otimes 4}, b_j^{\otimes 4} \rangle$ This is commonly used objective function for decomposing tensor $\sum_i a_i^\star b_i^{\star \otimes 4}$. It's known that for orthogonal $b_i^\star$'s, the only local minima are $\pm b_1^\star, \ldots, \pm b_d^\star$ Ge et al. (2015). Therefore, intuitively $R(B)$ pushes each of the $b_i$'s towards one of the $b_j^\star$'s. [10] Choosing $\mu$ to be small enough, it turns out that $P'(B) + R(B)$ doesn't have any spurious local minimum as we will show in Section B.

Another issue with the choice of $P'(B) + R(B)$ is that we are still having a constraint minimization problem. Such row-wise norm constraints only make sense when the ground-truth $B^\star$ is orthogonal and thus has unit row norm. A straightforward generalization of $P(B)$ to non-orthogonal case requires some special constraints that also depend on the covariance matrix $B^\star B^{\star \top}$, which in turn requires a specialized procedure to estimate. Instead, we move the constraints into the objective function by considering adding another regularization term that approximately enforces the constraints.

It turns out the following regularizer suffices for the orthogonal case: $S(B) = \lambda \sum_i (\|b_i\|^2 - 1)^2$. Moreover, we can extend this easily to the non-orthogonal case (see Section C) without estimating any statistics of $B^\star$ in advance. We note that $S(B)$ is not the Lagrangian multiplier and it does change the global minima slightly. We will take $\lambda$ to be large enough so that $\|b_i\|$ has to be close to 1. As a summary, we finally use the unconstrained objective

$$\min G(B) \triangleq P'(B) + R(B) + S(B)$$

Since $R(B)$ and $S(B)$ are degree-4 polynomials of $B$, the analysis of $G(B)$ is much more delicate, and we cannot use much linear algebra as we could for $P'(B)$. See Section B for details.

Finally we note that a feature of this objective $G(\cdot)$ is that it only takes $B$ as variables. We will estimate the value of $a^\star$ after we recover the value of $B$. (see Section D).

## 4    SIMULATION

In this section, we provide simple simulation results that verify that minimizing $G(B)$ with SGD recovers a permutation of $B^\star$; however, minimizing Equation (2.2) with SGD results in finding spurious local minima. Based on the formula for the population risk in Equation (2.3), we also

---

[10] However, note that $R(B)$ by itself doesn't work because it does not prevent the solutions where all the $b_i$'s are equal to the same $b_j^\star$.

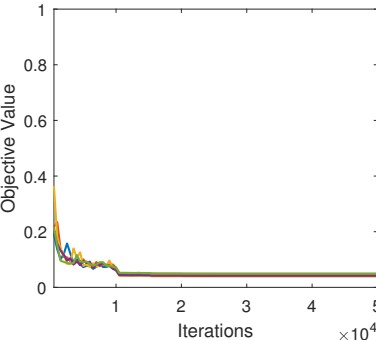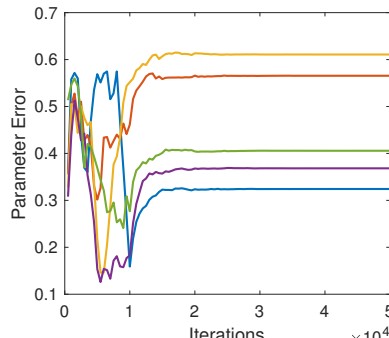

Figure 2: Data are generated by a network with sigmoid activation without noise. The training model uses the same architecture. Left: the estimated population risk doesn't converge to zero. Right: the parameter error using the surrogate in equation (4.1).

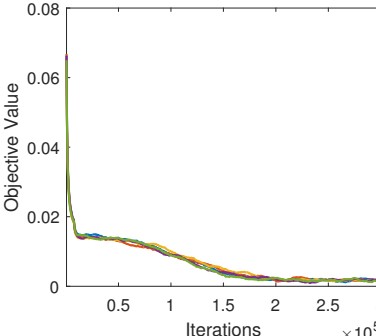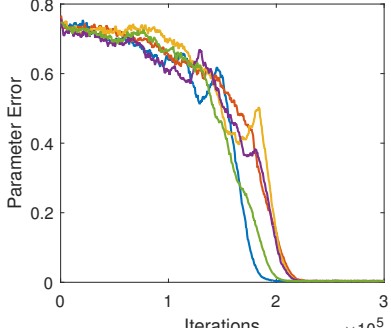

Figure 3: The labels are generated from a network with ReLU activation. We learn with $\hat{\sigma}_2 h_2 + \hat{\sigma}_4 h_4$ activation. Left: the test loss subtracted by the theoretical global minimum value. Right: the error in parameter space measured by equation (4.1)

verified empirically the conjecture that SGD would successfully recover $B^\star$ using the activation functions $\gamma(z) = \hat{\sigma}_2 h_2(z) + \hat{\sigma}_4 h_4(z)$,[11] even if the data were generated via a model with ReLU activation. (See Section 2.1 for the rationale behind such conjectures.)

For all of our experiments, we chose $B^\star = \mathrm{Id}_{d \times d}$ with dimension $d = 50$ and $a^\star = \mathbf{1}$ for simplicity, and the data is generated from a one-hidden-layer network with ReLU or Sigmoid activation without noise. We use stochastic gradient descent with fresh samples at each iteration, and we plot the (expected) population error (that is, the error on a fresh batch of examples).

To test whether SGD converges to a matrix $B$ which is equivalent to $B^\star$ up to permutation of rows, we use a surrogate error metric to evaluate whether $B^{\star-1}B$ is close to a permutation matrix. Given a matrix $Q$ with row norm 1, let

$$e(Q) = \min\{1 - \min_i \max_j |Q_{ij}|, 1 - \min_j \max_i |Q_{ij}|\}. \tag{4.1}$$

Then we have that if $e(Q) \leq \varepsilon$ for some $\varepsilon < 1/3$, then it implies that $Q$ is $\sqrt{2\varepsilon}$-close to a permutation matrix in infinity norm. On the other direction, we know that if $e(Q) > \varepsilon$, then $Q$ is not $\varepsilon$-close to any permutation matrix in infinity norm. The latter statement also holds when $Q$ doesn't have row norm 1.

Figure 1 shows that without over-parameterization, using ReLU as an activation function, SGD doesn't converge to zero test error and the ground-truth parameters. We decreased step-size by a

---

[11] We also observed that using $\gamma(z) = \frac{1}{2}|z|$ also works but due to the space limitation we don't report the experimental results here.

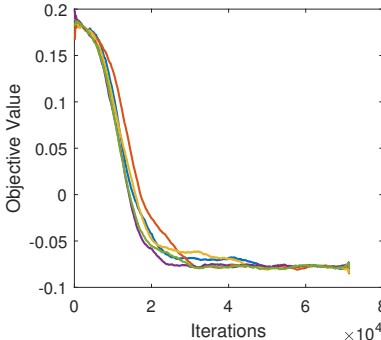 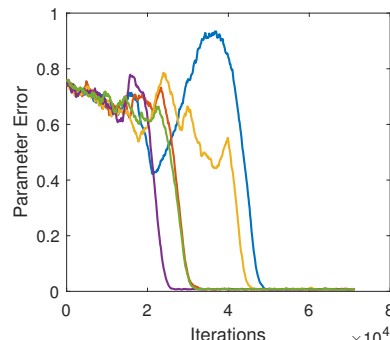

Figure 4: Learning with objective function $G(\cdot)$. Left: the test loss. Right: the error in parameter space measured by equation (4.1)

factor of $4$ every $5000$ number of iterations after the error plateaus at $10000$ iterations. For the final $5000$ iterations, the step-size is less than $10^{-9}$, so we can be confident that the non-zero objective value is not due to the variance of SGD. We see that none of the five runs of SGD converged to a global minimum. Figure 2 shows the result for sigmoid activation which is quantitatively similar.

Figure 3 shows that using $\hat{\sigma}_2 h_2 + \hat{\sigma}_4 h_4$ as the activation function, SGD with projection to the set of matrices $B$ with row norm 1 converges to the ground-truth parameters. We also plot the loss function which converges the value of a global minimum. (We subtracted the constant term in equation (2.6) so that the global minimum has loss 0.)

Figure 4 shows that using our objective function $G(B)$, the iterate converges to a permutation of the ground truth matrix $B^\star$. The fact that the parameter error goes up and down is not surprising, because the algorithm first gets close to a saddle point and then breaks ties and converges to a one of the global minima.

Finally we note that using the loss function $G(\cdot)$ seems to require significantly larger batch (and sample complexity) to reduce the variance in the gradients estimation. We used batch size $262144$ in the experiment for $G(\cdot)$. However, in contrast, for the $\hat{\sigma}_2 h_2 + \hat{\sigma}_4 h_4$ we used batch size $8192$ and for relu we used batch size $256$.

## 5 CONCLUSION

In this paper we first give an analytic formula for the population risk of the standard $\ell_2$ loss, which empirically may converge to a spurious local minimum. We then design a novel population loss that is guaranteed to have no spurious local minimum.

Designing objective functions with well-behaved landscape is an intriguing and potentially fruitful direction. We hope that our techniques can be useful for characterizing and designing the optimization landscape for other settings.

We conjecture that the objective $\alpha f_2 + \beta f_4$[12] has no spurious local minimum when $\alpha, \beta$ are reasonable constants and the ground-truth parameters are in general position. We provided empirical evidence to support the conjecture.

Our results assume that the input distribution is Gaussian. Extending them to other input distributions is a very interesting open problem.

ACKNOWLEDGEMENT

We thank Chi Jin for discussions at the beginning of this work. R.G. is funded by NSF CCF-1704860.

---

[12]See equation (2.4) for the definition of $f_k$ and Theorem 2.2 for how to access $\alpha f_2 + \beta f_4$ in the setting of one-hidden-layer neural nets.

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

# A  ANALYTIC FORMULA FOR POPULATION RISKS

## A.1  BASICS ON HERMITE POLYNOMIALS

In this section, we briefly review Hermite polynomials and Fourier analysis on Gaussian space. Let $H_j$ be the probabilists' Hermite polynomial Wikipedia (2017a), and let $h_j = \frac{1}{\sqrt{j!}} H_j$ be the normalized Hermite polynomials. The normalized Hermite polynomial forms a complete orthonormal basis in the function space $L^2(\mathbb{R}, e^{-x^2/2})$ in the following sense[13]. For two functions $f, g$ that map $\mathbb{R}$ to $\mathbb{R}$, define the inner product $\langle f, g \rangle$ with respect to the Gaussian measure as

$$\langle f, g \rangle = \mathop{\mathbb{E}}_{x \sim \mathcal{N}(0,1)} [f(x)g(x)] .$$

The polynomials $h_0, \ldots, h_m, \ldots$ are orthogonal to each other under this inner product:

$$\langle h_i, h_j \rangle = \delta_{ij} .$$

Here $\delta_{ij} = 1$ if $i = j$ and otherwise $\delta_{ij} = 0$. Given a function $\sigma \in L^2(\mathbb{R}, e^{-x^2/2})$, let the $k$-th Hermite coefficient of $\sigma$ be defined as

$$\hat{\sigma}_k = \langle \sigma, h_k \rangle .$$

Since $h_0, \ldots, h_m, \ldots$, forms a complete orthonormal basis, we have the expansion that

$$\sigma(x) = \sum_{k \in \mathbb{N}} \hat{\sigma}_k h_k(x) .$$

We will leverage several other nice properties of the Hermite polynomials in our proofs. The following claim connects the Hermite polynomial to the coefficients of Taylor expansion of a certain exponential function. It can also serve as a definition of Hermite polynomials.

---

[13]We denote by $L^2(\mathbb{R}, e^{-x^2/2})$ the weighted $L_2$ space, namely, $L^2(\mathbb{R}, e^{-x^2/2}) \triangleq \left\{ f : \int_{-\infty}^{\infty} f(x)^2 e^{-x^2/2} dx < \infty \right\}$

*Claim* A.1 ((O'Donnell, 2014, Equation 11.8)). We have that for $t, z \in \mathbb{R}$,

$$\exp(tz - \frac{1}{2}t^2) = \sum_{k=0}^{\infty} \frac{1}{k!} H_k(z) t^k .$$

The following Claims shows that the expectation $\mathbb{E}\left[h_n(x)h_m(y)\right]$ can be computed easily when $x, y$ are (correlated) Gaussian random variables.

*Claim* A.2 ((O'Donnell, 2014, Section 11.2)). Let $(x, y)$ be $\rho$-correlated standard normal variables (that is, both $x, y$ have marginal distribution $\mathcal{N}(0, 1)$ and $\mathbb{E}[xy] = \rho$). Then,

$$\mathbb{E}\left[h_m(x)h_n(y)\right] = \rho^n \delta_{mn} .$$

As a direct corollary, we can compute $\mathbb{E}_{x \sim \mathcal{N}(0, \mathrm{Id}_{d \times d})} \left[\sigma(u^\top x)\gamma(v^\top x)\right]$ by expanding in the Hermite basis and applying the Claim above.

*Claim* A.3. Let $\sigma, \gamma$ be two functions from $\mathbb{R}$ to $\mathbb{R}$ such that $\sigma^2, \gamma^2 \in L^2(\mathbb{R}, e^{-x^2/2})$. Then, for any unit vectors $u, v \in \mathbb{R}^d$, we have that

$$\underset{x \sim \mathcal{N}(0, \mathrm{Id}_{d \times d})}{\mathbb{E}} \left[\sigma(u^\top x)\gamma(v^\top x)\right] = \sum_{i \in \mathbb{N}} \hat{\sigma}_i \hat{\gamma}_i \langle u, v \rangle^i .$$

*Proof of Claim A.3.* Let $s = u^\top x$ and $t = v^\top x$. Then $s, t$ are two spherical standard normal random variables that are $\langle u, v \rangle$-correlated, and we have that

$$\underset{x \sim \mathcal{N}(0, \mathrm{Id}_{d \times d})}{\mathbb{E}} \left[\sigma(u^\top x)\gamma(v^\top x)\right] = \mathbb{E}\left[\sigma(s)\gamma(t)\right] .$$

We expand $\sigma(s)$ and $\gamma(t)$ in the Fourier basis and obtain that

$$\begin{aligned}
\mathbb{E}\left[\sigma(s)\gamma(t)\right] &= \mathbb{E}\left[\sum_{i \in \mathbb{N}} \hat{\sigma}_i h_i(s) \sum_{j \in \mathbb{N}} \hat{\gamma}_j h_j(t)\right] \\
&= \sum_{i,j} \hat{\sigma}_i \hat{\gamma}_j \, \mathbb{E}\left[h_i(s)h_j(t)\right] \\
&= \sum_i \hat{\sigma}_i \hat{\gamma}_i \langle u, v \rangle^i \qquad \qquad \text{(by Claim A.2)}
\end{aligned}$$

$\square$

## A.2 ANALYTIC FORMULA FOR POPULATION RISK $f$ AND $f'$

In this section we prove Theorem 2.1 and Theorem 2.2, which both follow from the following more general Theorem.

**Theorem A.4.** *Let* $\gamma, \sigma \in L^2(\mathbb{R}, e^{-x^2/2})$, *and* $\hat{y} = a^\top \gamma(Bx)$ *with parameter* $a \in \mathbb{R}^\ell$ *and* $B \in \mathbb{R}^{\ell \times d}$. *Define the population risk* $f_\gamma$ *as*

$$f_\gamma(a, B) = \mathbb{E}\left[\|y - \hat{y}\|^2\right] .$$

*Suppose* $B = \begin{bmatrix} b_1^\top \\ \vdots \\ b_\ell^\top \end{bmatrix}$ *and* $B^\star = \begin{bmatrix} b_1^{\star\top} \\ \vdots \\ b_m^{\star\top} \end{bmatrix}$ *and* $b_i$*'s and* $b_i^\star$*'s have unit* $\ell_2$ *norm. Then,*

$$f(a, B) = \sum_{k \in \mathbb{N}} \left\| \hat{\sigma}_k \sum_{i \in [m]} a_i^\star b_i^{\star \otimes k} - \hat{\gamma}_k \sum_{i \in [\ell]} a_i b_i^{\otimes k} \right\|_F^2 + \text{const},$$

*where* $\hat{\sigma}_k, \hat{\gamma}_k$ *are the* $k$*-th Hermite coefficients of the function* $\sigma$ *and* $\gamma$ *respectively.*

We can see that Theorem 2.1 follows from choosing $\gamma = \sigma$ and Theorem 2.2 follows from choosing $\gamma = \hat{\sigma}_2 h_2 + \hat{\sigma}_4 h_4$. The key intuition here is that we can decompose $\sigma$ into a weighted combination of Hermite polynomials, and each Hermite polynomial influence the population risk more or less independently (because they are orthogonal polynomials with respect to the Gaussian measure).

*Proof of Theorem A.4.* We have

$$
f_\gamma = \mathbb{E}\left[\|\hat{y} - y\|^2\right] = \mathbb{E}\left[\left\|a^{\star\top}\sigma(B^\star x) - a^\top\gamma(Bx)\right\|^2\right]
$$

$$
= \mathbb{E}\left[\left\|\sum_{i\in[m]} a_i^\star\sigma(b_i^{\star\top}x) - \sum_{i\in[\ell]} a_i\gamma(b_i^\top x)\right\|^2\right]
$$

$$
= \sum_{i\in[m],j\in[\ell]}\mathbb{E}\left[a_i^\star a_j^\star\sigma(b_i^{\star\top}x)\sigma(b_j^{\star\top}x)\right] + \sum_{i\in[m],j\in[\ell]}\mathbb{E}\left[a_ia_j\gamma(b_i^\top x)\gamma(b_j^\top x)\right]
$$

$$
- 2\sum_{i\in[m],j\in[\ell]}\mathbb{E}\left[a_i^\star a_j\sigma(b_i^{\star\top}x)\gamma(b_j^\top x)\right]
$$

$$
= \sum_{i,j\in[m]} a_i^\star a_j^\star\sum_{k\in\mathbb{N}}\hat{\sigma}_k^2\langle b_i^\star, b_j^\star\rangle^k + \sum_{i,j\in[\ell]} a_ia_j\sum_{k\in\mathbb{N}}\hat{\gamma}_k^2\langle b_i, b_j\rangle^k
$$

$$
- 2\sum_{i\in[m],j\in[\ell]} a_i^\star a_j\sum_{k\in\mathbb{N}}\hat{\sigma}_k\hat{\gamma}_k\langle b_i^\star, b_j\rangle^k \qquad\text{(by Claim A.3)}
$$

$$
= \sum_{k\in\mathbb{N}}\left\|\hat{\sigma}_k\sum_{i\in[m]} a_i^\star b_i^{\star\otimes k} - \hat{\gamma}_k\sum_{i\in[\ell]} a_i b_i^{\otimes k}\right\|_F^2 .
$$

$\square$

## A.3 ANALYTIC FORMULA FOR POPULATION RISK $G$

In this section we show that the population risk $G(\cdot)$ (defined as in equation (2.7)) has the following analytical formula:

$$
G(B) = 2\sqrt{6}|\hat{\sigma}_4|\cdot\sum_{i\in[d]} a_i^\star\sum_{j,k\in[d],j\neq k}\langle b_i^\star, b_j\rangle^2\langle b_i^\star, b_k\rangle^2
$$

$$
- \frac{|\hat{\sigma}_4|\mu}{\sqrt{6}}\sum_{i,j\in[d]} a_i^\star\langle b_i^\star, b_j\rangle^4 + \lambda\sum_{i=1}^m(\|b_i\|^2 - 1)^2 .
$$

The formula will be crucial for the analysis of the landscape of $G(\cdot)$ in Section B. The formula follows straightforwardly from the following two theorems and the definition (2.7).

**Theorem A.5.** *Let $\phi(\cdot,\cdot,\cdot)$ be defined as in equation (2.9), we have that*

$$
\mathbb{E}\left[y\cdot\sum_{j,k\in[d],j\neq k}\phi(b_j, b_k, x)\right] = 2\sqrt{6}\hat{\sigma}\cdot\sum_{i\in[d]} a_i^\star\sum_{j,k\in[d],j\neq k}\langle b_i^\star, b_j\rangle^2\langle b_i^\star, b_k\rangle^2 .
$$

**Theorem A.6.** *Let $\varphi(\cdot,\cdot)$ be defined as in equation (2.8), then we have that*

$$
\mathbb{E}\left[y\cdot\sum_{j\in[d]}\varphi(b_j, x)\right] = \frac{\hat{\sigma}_4}{\sqrt{6}}\sum_{i,j\in[d]} a_i^\star\langle b_i^\star, b_j\rangle^4 .
$$

In the rest of the section we prove Theorem A.5 and A.6.

We start with a simple but fundamental lemma. Essentially all the result in this section follows from expanding the two sides of equation (A.1) below.

**Lemma A.7.** *Let $u, v \in \mathbb{R}^d$ be two fixed vectors and $x \sim \mathcal{N}(0, \text{Id}_{d \times d})$. Then, for any $s, t \in \mathbb{R}$,*

$$\exp(\langle u, v \rangle st) = \mathbb{E}\left[\exp(u^\top xt - \frac{1}{2}\|u\|^2 t^2)\exp(v^\top xs - \frac{1}{2}\|v\|^2 s^2)\right]. \tag{A.1}$$

*Proof.* Using the fact that $\mathbb{E}\left[\exp(v^\top x)\right] = \exp(\frac{1}{2}\|v\|^2)$, we have that,

$$\mathbb{E}\left[\exp(u^\top xt - \frac{1}{2}\|u\|^2 t^2)\exp(v^\top xs - \frac{1}{2}\|v\|^2 s^2)\right]$$

$$= \mathbb{E}\left[\exp((tu + sv)^\top x)\right]\exp(-\frac{1}{2}\|u\|^2 t^2 - \frac{1}{2}\|v\|^2 s^2)$$

$$= \exp(\frac{1}{2}\|tu + sv\|^2 - \frac{1}{2}\|u\|t^2 - \frac{1}{2}\|v\|^2 s^2) \quad \text{(by the formula } \mathbb{E}\left[\exp(v^\top x)\right] = \exp(\frac{1}{2}\|v\|^2))$$

$$= \exp(\langle u, v \rangle st).$$

$\square$

Next we extend some of the results in the previous section to the setting with different scaling (such as when $v$ in Claim A.3 is no longer a unit vector.)

**Lemma A.8.** *Let $u$ be a fixed unit vector and $v$ be an arbitrary vector in $\mathbb{R}^d$. Let $\varphi(v, x) = \frac{1}{8}\|v\|^4 - \frac{1}{4}(v^\top x)^2\|v\|^2 + \frac{1}{24}(v^\top x)^4$.*

$$\langle u, v \rangle^4 \delta_{4,k} = \mathbb{E}\left[H_k(u^\top x)\varphi(v, x)\right] \tag{A.2}$$

As a sanity check, we can verify that when $v$ is a unit vector, $\varphi(v, x) = \sqrt{24}h_4(v^\top x)$ and th Lemma reduces to a special case of Claim A.2.

*Proof.* Let $A, B$ be formal power series in variable $s, t$ defined as $A = \exp(\langle u, v \rangle st)$ and $B = \mathbb{E}\left[\exp(u^\top xt - \frac{1}{2}\|u\|^2 t^2)\exp(v^\top xs - \frac{1}{2}\|v\|^2 s^2)\right]$. We refer the readers to Wikipedia (2017b) for more backgrounds of power series. For casual readers, one can just think of $A$ as $B$ as two power series obtained by expanding the $exp(\cdot)$ via Taylor expansion. For a formal power series $A$ in variable $x$, let $[x^\alpha]A$ to denote coefficient in front of the monomial $x^\alpha$. By Lemma A.7, we have that $A = B$, and thus

$$[s^4 t^k]A = [s^4 t^k]B, \tag{A.3}$$

which implies that

$$\frac{1}{24}\langle u, v \rangle^4 \delta_{4,k} = \mathbb{E}\left[[t^k]\left(\exp(u^\top xt - \frac{1}{2}t^2)\right) \cdot [s^4]\left(\exp(v^\top xs - \frac{1}{2}\|v\|^2 s^2)\right)\right]$$

$$= \mathbb{E}\left[\frac{1}{k!}H_k(u^\top x)\varphi(v, x)\right]. \tag{A.4}$$

where the last line is by the fact that $\varphi(v, x) = [s^4]\exp(v^\top xs - \frac{1}{2}\|v\|^2 s^2)$. This can be verified by applying Claim A.1 with $t = s\|v\|$ and $z = \frac{v^\top x}{\|v\|}$, and noting that $H_4(x) = x^4 - 6x^2 + 3$. $\square$

Now we are ready prove Theorem A.6 using Lemma A.8.

*Proof of Theorem A.6.* Using the fact that $\sigma(v^\top x) = \sum_{k=0}^\infty \hat{\sigma}_k h_k(v^\top x)$, we have that

$$\mathbb{E}\left[y \cdot \sum_j \varphi(b_j, x)\right] = \sum_{i,j \in [d]} a_i^\star \mathbb{E}\left[\sigma(b_i^{\star\top} x)\varphi(b_j, x)\right]$$

$$= \sum_{i,j \in [d]} a_i^\star \sum_k^\infty \mathbb{E}\left[\hat{\sigma}_k h_k(b_i^{\star\top} x)\varphi(b_j, x)\right]$$

$$= \sum_{i,j\in[d]} a_i^\star \, \mathbb{E}\left[\hat{\sigma}_4 h_4(b_i^{\star\top}x)\varphi(b_j,x)\right] = \frac{\hat{\sigma}_4}{\sqrt{6}} \sum_{i,j\in[d]} a_i^\star \langle b_i^\star, b_j\rangle^4$$

$$\text{(by Lemma A.8 and } h_j = \tfrac{1}{\sqrt{j}}H_j)$$

$\square$

Towards proving Theorem A.5, we start with the following Lemma. Inspired by the proofs above, we design a function $\phi(v,w,x)$ such that we can estimate $\langle u,v\rangle^2\langle u,w\rangle^2$ by taking expectation of $\mathbb{E}\left[\sigma(u^\top x)\phi(v,w,x)\right]$.

**Lemma A.9.** *Let $a$ be a fixed unit vector in $\mathbb{R}^d$ and $v,w$ two fixed vectors in $\mathbb{R}^d$. Let $\varphi(\cdot,\cdot)$ be defined as in Lemma A.8. Define $\phi(v,w,x)$ as*

$$\phi(v,w,x) = \varphi(v+w,x) + \varphi(v-w,x) - 2\varphi(v,x) - 2\varphi(w,x) \tag{A.5}$$

$$= \frac{1}{2}\|v\|^2\|w\|^2 + \langle v,w\rangle^2 - \frac{1}{2}\|w\|^2(v^\top x)^2 - \frac{1}{2}\|v\|^2(w^\top x)^2 \tag{A.6}$$

$$- 2(v^\top x)(w^\top x)v^\top w + \frac{1}{2}(v^\top x)^2(w^\top x)^2 \,.$$

*Then, we have that*

$$\mathbb{E}\left[\sigma(u^\top x)\phi(v,w,x)\right] = 2\sqrt{6}\hat{\sigma}_4\langle u,v\rangle^2\langle u,w\rangle^2 \,.$$

*Proof.* Using the fact that $\langle u,v+w\rangle^2 + \langle u,v-w\rangle^4 - 2\langle u,v\rangle^2 - 2\langle u,w\rangle^4 = 12\langle u,v\rangle^2\langle u,w\rangle^2$ and Lemma A.8, we have that

$$12\langle u,v\rangle^2\langle u,w\rangle^2\delta_{4,k} = \mathbb{E}\left[H_k(u^\top x)(\varphi(v+w,x) + \varphi(v-w,x) - 2\varphi(v,x) - 2\varphi(w,x))\right]$$

$$= \mathbb{E}\left[H_k(u^\top x)\phi(v,w,x)\right] \,. \tag{A.7}$$

Using the fact that $\sigma(u^\top x) = \sum_{k=0}^\infty \hat{\sigma}_k h_k(u^\top x)$, we conclude that

$$\mathbb{E}\left[\sigma(u^\top x)\phi(v,w,x)\right] = \sum_{k=0}^\infty \hat{\sigma}_k \, \mathbb{E}\left[h_k(u^\top x)\phi(v,w,x)\right]$$

$$= \frac{\hat{\sigma}_4}{\sqrt{6}} \mathbb{E}\left[H_4(u^\top x)\phi(v,w,x)\right] \qquad \text{(by Lemma A.8 and } h_j = \tfrac{1}{\sqrt{j}}H_j)$$

$$= 2\sqrt{6}\hat{\sigma}_4\langle u,v\rangle^2\langle u,w\rangle^2 \qquad\qquad \text{(by Lemma A.8 again)}$$

$\square$

Now we are ready to prove Theorem A.5 by using Lemma A.9 for every summand.

*Proof of Theorem A.5.* We have that

$$\mathbb{E}\left[y\cdot\sum_{j,k}\phi(b_j,b_k,x)\right] = \sum_i a_i^\star \sum_{j,k}\mathbb{E}\left[\sigma(b_i^{\star\top}x)\phi(b_j,b_k,x)\right]$$

$$= 2\sqrt{6}\hat{\sigma}_4\sum_i a_i^\star \sum_{j,k}\langle b_i^\star, b_j\rangle^2\langle b_i^\star, b_k\rangle^2 \,. \qquad \text{(by Lemma A.9)}$$

$\square$

# B  LANDSCAPE OF POPULATION RISK $G(\cdot)$

In this section we prove Theorem 2.3. Since the landscape property is invariant with respect to rotations of parameters, without loss of generality we assume $B^\star$ is the identity matrix Id throughout

this section. (See Section C for a precise statement for the invariance.) Recall that by Theorem 2.6, the population risk $G(\cdot)$ in the case of $B^\star = \text{Id}$ is equal to

$$G(B) = 2\sqrt{6}|\hat{\sigma}_4| \sum_i a_i^\star \sum_{j \neq k} (b_j^\top e_i)^2 (b_k^\top e_i)^2$$

$$- \frac{|\hat{\sigma}_4|\mu}{\sqrt{6}} \sum_{i=1}^d a_i^\star \sum_{j=1}^d (b_j^\top e_i)^4 + \lambda \sum_{j=1}^d \left( \|b_j\|^2 - 1 \right)^2. \tag{B.1}$$

In the rest of section we work with the formula above for $G(\cdot)$ instead of the original definition. In fact, for future reference, we study a more general version of the function $G$. For nonnegative vectors $\alpha, \beta$ and nonnegative number $\mu$, let $G_{\alpha,\beta,\mu}$ be defined as

$$G_{\alpha,\beta,\mu}(B) = \sum_{i=1}^d \alpha_i \sum_{j \neq k} (b_j^\top e_i)^2 (b_k^\top e_i)^2 - \mu \sum_{i=1}^d \beta_i \sum_{j=1}^d (b_j^\top e_i)^4 + \lambda \sum_{j=1}^d (\|b_j\|^2 - 1)^2 \tag{B.2}$$

Here $e_i$ denotes the $i$-th natural basis vector. We see that $G$ is sub-case of $G_{\alpha,\beta,\mu}$ and we prove the following extension of Theorem 2.3. Let $\alpha_{\max} = \max_i \alpha_i$ and $\alpha_{\min} = \min_i \alpha_i$.

**Theorem B.1.** *Let $\kappa_\alpha = \alpha_{\max}/\alpha_{\min}$ and $c$ be a sufficiently small universal constant (e.g. $c = 10^{-2}$ suffices). Suppose $\mu \leq c\alpha_{\min}/\beta_{\max}$ and $\lambda \geq 4\max(\mu\beta_{\max}, \alpha_{\max})$. Then, the function $G_{\alpha,\beta,\mu}(B)$ defined as in equation* (B.2) *satisfies that*

1. *A matrix $B$ is a local minimum of $G_{\alpha,\beta,\mu}$ if and only if $B$ can be written as $B = DP$ where $P$ is a permutation matrix and $D$ is a diagonal matrix with $D_{ii} \in \left\{ \pm\sqrt{\frac{1}{1-\mu\beta_i/\lambda}} \right\}$.*

2. *Any saddle point $B$ has strictly negative curvature in the sense that $\lambda_{\min}(\nabla^2 G_{\alpha,\beta,\mu}(B)) \leq -\tau_0$ where $\tau_0 = c\min\{\mu\beta_{\min}/(\kappa_\alpha d), \mu\beta_{\min}^2/\beta_{\max}, \lambda\}$*

3. *Suppose $B$ is an approximate local minimum in the sense that $B$ satisfies*

$$\|\nabla g(B)\| \leq \varepsilon \text{ and } \lambda_{\min}(\nabla^2 g(B)) \geq -\tau_0$$

   *Then $B$ can be written as $B = DP + E$ where $P$ is a permutation matrix, $D$ is a diagonal matrix with the entries satisfying*

$$\frac{1}{1 - \frac{\mu\beta_i}{\lambda}} \left( 1 - \frac{18d\varepsilon^2}{\beta_{\min}^2} - \frac{\varepsilon}{2\lambda} \right) \leq D_{ii}^2 \leq \frac{1}{1 - \frac{\mu\beta_i}{\lambda}} \left( 1 + \frac{\varepsilon}{2\lambda} \right)$$

   *and $E$ is an error matrix satisfying*

$$|E|_\infty \leq 3\varepsilon/\beta_{\min}.$$

   *As a direct consequence, $B$ is $O_d(\varepsilon)$-close to a global minimum in Euclidean distance, where $O_d(\cdot)$ hides polynomial dependency on $d$ and other parameters.*

Here we recall that $|E|_\infty$ denotes the largest entries in the matrix $E$. Theorem 2.3 follows straightforwardly from Theorem B.1 by setting $\alpha = 2\sqrt{6}|\hat{\sigma}_4|a^\star$ and $\beta = |\hat{\sigma}_4|a^\star/\sqrt{6}$. In the rest of the section we prove Theorem B.1.

Note that our variable $B$ is a matrix of dimension $d \times d$ and we use $b_i$ to denote the rows of $B$, that is, $B = \begin{bmatrix} b_1^\top \\ \vdots \\ b_d^\top \end{bmatrix}$. Naturally, towards analyzing the properties of a local minimum $B$, the first step is that we pick a row $b_s$ of $B$ and treat only $b_s$ as variables and others rows as fixed. We will show that local optimality of $b_s$ will imply that $b_s$ is equal to one of the basis vector $e_j$ up to some scaling factor. This step is done in Section B.1. Then in Section B.2 we show that the local optimality of all the variables in $B$ implies that each of the rows of $B$ corresponds to different basis vector, which implies that $B$ is a permutation matrix (up to scaling of the rows).

B.1   STEP 1: ANALYSIS OF LOCAL OPTIMALITY OF A SINGLE ROW

Suppose we fix $b_1, \cdots, b_{s-1}, b_{s+1}, \cdots, b_d$, and optimize only over $b_s$, we obtain the objective $h$ of the following form:

$$h_{\alpha,\beta,\lambda}(x) = \sum_{i=1}^{d} \alpha_i x_i^2 - \sum_{i=1}^{d} \beta_i x_i^4 + \lambda(\|x\|^2 - 1)^2 \tag{B.3}$$

We can see that setting $\alpha_i = a_i^\star \sum_{k \neq s}(b_k^\top e_i)^2$, $\beta_i = a_i^\star$, and $x = b_s$ gives us the original objective $G(B)$. In this subsection, we will work with $h(\cdot)$ and analyze the properties of the local minima of $h(\cdot)$.

The following lemma shows that a local minimum $x$ of the objective $h(\cdot)$ must be a scaling of a basis vector. For a vector $x$, let $|x|_{2\text{nd}}$ denotes the second largest absolute values of the entries for $x$. We note that $|\cdot|_{2\text{nd}}$ is not a norm. The lemma deals generally an approximate local minimum, though we suggest casual readers simply think of $\varepsilon, \tau = 0$ in the lemma.

**Lemma B.2.** *Let $h(\cdot)$ be defined in equation* (B.3) *with non-negative vectors $\alpha$ and $\beta$ in $\mathbb{R}^d$. Suppose parameters $\varepsilon, \tau \geq 0$ satisfy that $\varepsilon \leq \sqrt{\tau^3/\beta_{\min}}$. If some point $x$ satisfies $\|\nabla h(x)\| \leq \varepsilon$ and $\lambda_{\min}(\nabla^2 h(x)) \geq -\tau$, then we have*

$$|x|_{2\text{nd}} \leq \sqrt{\frac{\tau}{\beta_{\min}}}.$$

*Proof.* Without loss of generality, we can take $\varepsilon = \sqrt{\tau^3/\beta_{\min}}$ which means $\tau = \varepsilon^{2/3}\beta_{\min}^{1/3}$. The gradient and Hessian of function $h(\cdot)$ are

$$\nabla h(x) = 2\operatorname{diag}(\alpha)x - 4\operatorname{diag}(\beta)x^{\odot 3} + \gamma x$$
$$\nabla^2 h(x) = 2\operatorname{diag}(\alpha) - 12\operatorname{diag}(\beta \odot x^{\odot 2}) + \gamma\operatorname{Id} + 8\lambda xx^\top. \tag{B.4}$$

where $\gamma \triangleq 4\lambda(\|x\|^2 - 1)$.

Let $S = \{i : |x_i| \geq \delta\}$ be the indices of the coordinates that are significantly away from zero, where $\delta = \left(\frac{\varepsilon}{\beta_{\min}}\right)^{1/3}$. Since $\|\nabla h(x)\| \leq \varepsilon$, we have that $|\nabla h(x)_i| \leq \varepsilon$ for every $i \in [d]$, which implies that

$$\forall i \in [d], \left|2\alpha_i x_i + \gamma x_i - 4\beta_i x_i^3\right| \leq \varepsilon \tag{B.5}$$

which further implies that

$$\forall i \in S, \left|2\alpha_i + \gamma - 4\beta_i x_i^2\right| \leq \frac{\varepsilon}{\delta} \tag{B.6}$$

If $|S| = 1$, then we are done because $|x|_{2\text{nd}} \leq \delta$. Next we prove that $|S| \geq 2$. For the sake of contradiction, we assume that $|S| \geq 2$. Moreover, WLOG, we assume that $|x|_1 \geq |x|_2$ are the two largest entries of $|x|$ in absolute values.

We take $v \in \mathbb{R}^d$ such that $v_1 = -x_2/\sqrt{x_1^2 + x_2^2}$, and $v_2 = x_1/\sqrt{x_1^2 + x_2^2}$, and $v_j = 0$ for $j \geq 2$. Then we have that $v^\top x = 0$ and $\|v\| = 1$. We evaluate the quadratic form and have that

$$\begin{aligned}
v^\top \nabla^2 h(x) v &= v^\top (2\operatorname{diag}(\alpha) + \gamma I_d)v - 12 v^\top \operatorname{diag}(\beta \odot x^{\odot 2})v && \text{(since } v^\top x = 0) \\
&= (2\alpha_1 + \gamma)v_1^2 + (2\alpha_2 + \gamma)v_2^2 - 12\beta_1 v_1^2 x_1^2 - 12\beta_2 v_2^2 x_2^2 \\
&\leq -8\beta_1 v_1^2 x_1^2 - 8\beta_2 v_2^2 x_2^2 + \frac{\varepsilon}{\delta} && \text{(by equation (B.6) and } \|v\| = 1) \\
&\leq -8(\beta_1 + \beta_2)\frac{x_2^2 x_1^2}{x_1^2 + x_2^2} + \frac{\varepsilon}{\delta} \\
&\leq -8\beta_{\min} x_2^2 + \frac{\varepsilon}{\delta}.
\end{aligned}$$

Recall that $\delta = \left(\frac{\varepsilon}{\beta_{\min}}\right)^{1/3}$. Then we conclude that

$$v^\top \nabla^2 h(x)v \leq -6\beta_{\min}^{1/3}\varepsilon^{2/3} = -6\tau.$$

This contradicts with the assumption that $\lambda_{\min}(\nabla^2 h(x)) \geq -\beta_{\min}^{1/3}\varepsilon^{2/3} = \tau$ and that $\|v\| = 1$. Therefore we have $|S| = 1$ and

$$|x|_{2\text{nd}} \leq \delta = \left(\frac{\varepsilon}{\beta_{\min}}\right)^{1/3} \leq \sqrt{\frac{\tau}{\beta_{\min}}} \qquad \text{(using } \varepsilon \leq \sqrt{\tau^3/\beta_{\min}}\text{)}$$

$\square$

For future reference, we can also show that for a sufficiently strong regularization term (sufficiently large $\lambda$), the norm of a local minimum $x$ should be bounded from below and above by $1/2$ and $2$. This are rather coarse bounds that suffice for our purpose in this subsection. In Section B.2 we will show that all the rows of a local minimum $B$ of $G$ have norm close to 1.

**Lemma B.3.** *In the setting of Lemma B.2,*

1. *Suppose in addition that $\lambda \geq 4\max(\beta_{\max}, \tau)$ and $\varepsilon \leq 0.1\beta_{\min}d^{-3/2}$, then*

$$\|x\|^2 \leq 2\,.$$

2. *Let $i^\star = \arg\max_i |x_i|$. In addition to the previous conditions in bullet 1, assume that $\lambda \geq 4\alpha_{i^\star}$. Then,*

$$\|x\|^2 \geq \frac{1}{2}\,.$$

We remark that we have to state the conditions for the upperbounds and lowerbounds separately since they will be used with these different conditions.

*Proof.* Let $S = \{i : |x_i| \geq \delta\}$ be the indices of the coordinates that are significantly away from zero, where $\delta = \left(\frac{\varepsilon}{\beta_{\min}}\right)^{1/3}$. We first show that $\|x\|^2 \leq 2$. We divide into two cases:

1. $S$ is empty. Since $\varepsilon \leq 0.1\beta_{\min}d^{-3/2}$, then $\delta \leq \frac{\sqrt{2}}{\sqrt{d}}$. We conclude that $\|x\|^2 \leq 2$.

2. $S$ is non-empty. For $i \in S$, recall equation (B.6) which implies that

$$4\lambda(\|x\|^2 - 1) \leq \frac{\varepsilon}{\delta} + 4\beta_i x_i^2$$
$$\leq \frac{\varepsilon}{\delta} + 4\beta_{\max}\|x\|^2$$
$$\|x\|^2 \leq \frac{\varepsilon}{4\lambda\delta} + \frac{\beta_{\max}}{\lambda}\|x\|^2 + 1$$

Since $\lambda \geq 4\beta_{\max}$, so $\lambda \geq \beta_{\min}^{1/3}\varepsilon^{2/3} \geq \frac{3\varepsilon}{4\delta}$, and thus from the display above we have that $\|x\|^2 \leq 2$.

Next we show that $\|x\|^2 \geq \frac{1}{2}$. Again we divide into two cases:

1. $S$ is empty. For the sake of contradiction, assume that $\|x\|^2 \leq \frac{1}{2}$, then $\gamma \leq -2\lambda$. We show that there is sufficient negative curvature. Recall that

$$\nabla^2 h(x) = 2\operatorname{diag}(\alpha) - 12\operatorname{diag}(\beta \odot x^{\odot 2}) + \gamma I + 8\lambda xx^\top$$
$$\preceq 2\operatorname{diag}(\alpha) - 12\operatorname{diag}(\beta \odot x^{\odot 2}) - 2\lambda I + 8\lambda xx^\top$$

Choose index $j^\star$ so that $\alpha_{j^\star} = \alpha_{\min}$, then

$$e_{j^\star}^\top \nabla^2 h(x) e_{j^\star} = 2\alpha_{\min} - 12\beta_{j^\star}x_{j^\star}^2 - 2\lambda + 8\lambda x_{j^\star}^2$$
$$\leq 2\alpha_{\min} + 8\lambda\delta^2 - 2l$$
$$\leq 2\alpha_{\min} - \lambda(2 - 8\delta^2)$$

$$\leq 2\alpha_{\min} - \frac{4}{3}\lambda \qquad\qquad \text{(by } \delta^2 \leq \tfrac{1}{12})$$

$$\leq -\frac{5}{6}\lambda \leq -3\tau \qquad\qquad \text{(by } \lambda \geq 4\max\{\alpha_{\min}, \tau\})$$

This contradicts with the fact that $\lambda_{\min}(\nabla^2 h(x)) \geq -\tau$. Thus when $S$ is empty, $\|x\|^2 \geq \frac{1}{2}$.

2. $S$ is non-empty. Recall that $i^\star = \arg\max_i |x_i|$, and by definition $i^\star \in S$. Using Equation (B.6)

$$\gamma \geq -2\alpha_{i^\star} - \frac{\varepsilon}{\delta}$$

which implies that

$$\|x\|^2 \geq 1 - \frac{\alpha_{i^\star}}{\lambda} - \frac{\varepsilon}{4\lambda\delta}.$$

Since $\lambda \geq 4\alpha_{i^\star}$, and $\lambda \geq \beta_{\min}^{1/3}\varepsilon^{2/3} \geq \frac{\varepsilon}{\delta}$, we conclude that $\|x\|^2 \geq 1/2$.

$$\square$$

We have shown that a local minimum $x$ of $h$ should be a scaling of the basis vector $e_{i^\star}$. The following lemma strengthens the result by demonstrating that not all basis vector can be a local minimum — the corresponding coefficient $\alpha_{i^\star}$ has to be reasonably small for $e_{i^\star}$ being a local minimum. The key intuition here is that if $\alpha_{i^\star}$ is very large compared to other entries of $\alpha$, then if we move locally the mass of $e_{i^\star}$ from entry $i^\star$ to some other index $j$, the objective function will be likely to decrease because $\alpha_j x_j^2$ is likely to be smaller than $\alpha_{i^\star} x_{i^\star}^2$. (Indeed, we will show that such movement will cause a second-order decrease of the objective function in the proof.)

**Lemma B.4.** *In the setting of Lemma B.2, let* $i^\star = \arg\max_i |x_i|$. *If* $\|\nabla h(x)\| \leq \varepsilon$, *and* $\lambda_{\min}(\nabla^2 h(x)) > -\tau$ *for* $0 \leq \tau \leq 0.1\beta_{\min}/d$ *and* $\varepsilon \leq \sqrt{\tau^3/\beta_{\min}}$, *then*

$$\alpha_{i^\star} \leq \alpha_{\min} + 2\varepsilon + 2\tau + 4\beta_{i^\star}.$$

*Proof.* For the ease of notation, assume WLOG that $i^\star = 1$. Let $\delta = (\tau/\beta_{\min})^{1/2}$. By the assumptions, we have that $\delta \leq \frac{1}{\sqrt{6d}}$. By Lemma B.2, we have $\|x\|^2 \geq \frac{1}{2}$, which implies that

$$x_1^2 \geq \|x\|^2 - (d-1)|x|_{2\text{nd}}^2 \geq \frac{1}{2} - d|x|_{2\text{nd}}^2 \geq 1 - d\delta^2 \geq \frac{1}{3}. \qquad (B.7)$$

Define $v = -\left(\frac{x_k}{x_1}\right)e_1 + e_k$. Since $x_1$ is the largest entry of $x$, we can verify that $1 \leq \|v\|^2 = 1 + \frac{x_k^2}{x_1^2} \leq 2$. By the assumption, we have that

$$v^\top \nabla^2 h(x) v \geq -\tau \|v\|^2 \geq -4\tau. \qquad (B.8)$$

On the other hand, recall the form of Hessian (equation (B.4)), by straightforward algebraic manipulation, we have that

$$v^\top \nabla^2 h(x) v = v^\top \left(2\,\mathrm{diag}(\alpha) - 12\,\mathrm{diag}(\beta \odot x^{\odot 2}) + \gamma\mathrm{Id} + 8\lambda xx^\top.\right) v$$

$$= 2\alpha_1\left(\frac{x_k}{x_1}\right)^2 + 2\alpha_k - 12\beta_1 x_k^2 - 12\beta_k x_k^2 + \gamma\left(\frac{x_k}{x_1}\right)^2 + \gamma \qquad \text{(by } v^\top x = 0)$$

$$\leq (2\alpha_1 + \gamma)\left(\frac{x_k}{x_1}\right)^2 - 12(\beta_1 + \beta_k)x_k^2 + 2\alpha_k + \left(4\beta_1 x_1^2 - 2\alpha_1 + \frac{\varepsilon}{|x_1|}\right)$$

$$\qquad\qquad\qquad\qquad \text{(by equation (B.6))}$$

$$\leq \left(4\beta_1 x_1^2 + \frac{\varepsilon}{|x_1|}\right)\left(\frac{x_k}{x_1}\right)^2 - 12(\beta_1 + \beta_k)x_k^2 + 2\alpha_k + \left(4\beta_1 x_1^2 - 2\alpha_1 + \frac{\varepsilon}{|x_1|}\right)$$

$$\qquad\qquad\qquad\qquad \text{(by equation (B.6))}$$

$$= -8\beta_1 x_k^2 - 12\beta_k x_k^2 + 4\beta_1 x_1^2 + 2\alpha_k - 2\alpha_1 + 4\varepsilon$$
$$\text{(by } |x_k| \leq |x_1| \text{ and } |x_1|^2 \geq 1/3)$$
$$\leq 2\alpha_k - 2\alpha_1 + 4\varepsilon + 8\beta_1 . \qquad \text{(by } \|x\| \leq 2 \text{ using Lemma B.2)}$$

Combining equation (B.8) and the equation above gives

$$\alpha_1 \leq \alpha_k + 2\varepsilon + 2\tau + 4\beta_1 .$$

Since $k$ is arbitrary we complete the proof. $\qquad\square$

The previous lemma implies that it's very likely that the local minimum $x$ can be written as $x = x_{i^\star} e_{i^\star}$ and the index $i^\star$ is also likely to be the argmin of $\alpha$. The following technical lemma shows that when this indeed happens, then we can strengthen Lemma B.2 in terms of the error bound's dependency on $\varepsilon$ and $\tau$. In Lemma B.2, we have that $|x|_{2nd}$ is bounded by a function of $\tau$. Here we strengthen the bound to be a function that only depends on $\varepsilon$. Thus as long as $\tau$ be small enough so that we can apply Lemma B.2 and Lemma B.4 to meet the condition of the lemma below, then we get an error bound that goes to zero as $\varepsilon$ goes to zero. This translates to the error bound in bullet 3 of Theorem B.1 where the bound on $E$ only depends on $\varepsilon$. For casual readers we suggest to skip this Lemma since its precise functionality will only be clearer in the proof of Theorem B.1.

**Lemma B.5.** *In the setting of Lemma B.2, in addition we assume that $i = \operatorname{argmin}_k |\alpha_k|$ and that $x$ can be written as $x = x_i e_i + x_{-i}$ satisfying*

$$\|x_{-i}\|_\infty \leq 0.1 \min\{1/\sqrt{d}, \sqrt{\beta_{\min}/(\beta_{\max})}\} .$$

*Then, we can strengthen the bound to*

$$\|x_{-i}\|_\infty \leq \frac{3\varepsilon}{\beta_{\min}} .$$

*Proof.* WLOG, let $i = 1$. Let $x_j$ be the second largest entry of $x$ in absolute value. Define $v_1 = 4\beta_1 x_1^2 - 2\alpha_1 - \gamma$, and similarly $v_j = 4\beta_j x_j^2 - 2\alpha_j - \gamma$. Since $\|\nabla h(x)\| \leq \varepsilon$, by equation (B.6), we have that $|v_1| \leq \frac{\varepsilon}{|x_1|}$ and $|v_2| = \frac{\varepsilon}{|x_j|}$. Subtracting $4\beta_1 x_1^2 = 2\alpha_1 + \gamma + v_1$ and $4\beta_j x_j^2 = 2\alpha_j + \gamma + v_j$, we obtain,

$$4\beta_1 x_1^2 = 4\beta_j x_j^2 - 2(\alpha_j - \alpha_1) + v_1 - v_j$$
$$\leq 4\beta_j x_j^2 + (v_1 - v_j) \qquad \text{(since } \alpha_j - \alpha_1 \geq 0)$$

Since $\|x\|^2 \geq \frac{1}{2}$, then $x_1^2 \geq \frac{1}{2} - d\delta^2 \geq \frac{1}{3}$. Since $|x_j| \leq \delta$,

$$4\beta_j x_j^2 \leq 4\beta_{\max}\delta^2$$

Combining the above two displays,

$$(v_1 - v_j) \geq 4\beta_1 x_1^2 - 4\beta_j \delta^2$$
$$\geq 4\beta_1 x_1^2 - 4\beta_{\max}\delta^2$$
$$\geq \frac{4}{3}\beta_1 - \frac{2}{3}\beta_{\min} \qquad \text{(using } x_1^2 \geq \frac{1}{3} \text{ and } \delta \leq \sqrt{\beta_{\min}/(6\beta_{\max})})$$
$$\geq \frac{2}{3}\beta_{\min} \qquad\qquad\qquad\qquad\qquad (B.9)$$

Since $|v_1| \leq \frac{\varepsilon}{|x_1|}$ and $|v_2| = \frac{\varepsilon}{|x_2|}$,

$$2\frac{\varepsilon}{|x_j|} \geq \frac{2}{3}\beta_{\min}, \qquad\qquad\qquad\qquad (B.10)$$

and re-arranging gives $|x_j| \leq 3\frac{\varepsilon}{\beta_{\min}}$. $\qquad\square$

## B.2 Local Optimality of All the Variables

In this section we prove Theorem B.1. Results in Subsection B.1 have established that if $B$ is a local minimum, then each row $b_s$ of $B$ has to be a scaling of a basis vector. In this section we show that these basis vectors need to be distinct from each other. The following proposition summaries such a claim (with a weak error analysis).

**Proposition B.6.** *In the setting of Theorem B.1, suppose $B$ satisfies*

$$\|\nabla g(B)\| \leq \varepsilon \text{ and } \lambda_{\min}(\nabla^2 g(B)) \geq -\tau$$

*for parameters $\tau, \varepsilon$ satisfying $0 \leq \tau \leq c\min\{\mu\beta_{\min}/(\kappa_\alpha d), \lambda\}$ and $\varepsilon \leq c\min\{\alpha_{\min}, \sqrt{\tau^3/\beta_{\min}}\}$. Then, the matrix $B$ can be written as*

$$B = DP + E\,,$$

*where $D$ is diagonal such that $\forall i, |D_{ii}| \in [1/4, 2]$, and $P$ is a permutation matrix, and $|E|_\infty \leq \delta$ with $\delta = \left(\frac{\tau}{\mu\beta_{\min}}\right)^{1/2}$.*

As alluded before, in the proof we will first apply the results in Section B.1 to show that when $B$ is a local minimum, each row $b_s$ has a unique large entry. Then we will show that the largest entries of each row sit on different columns. The key intuition behind the proof is that if two rows, say row $s, t$, have their large entries on the same column, then it means that there exists a column— say column $k$ — that doesn't contain largest entry of any row. Then either row $s$ or $t$ will violate Lemma B.4. Or in other words, either row $s$ or $t$ can move their mass into the column $k$ to decrease the function value. This contradicts the assumption that $B$ is a local minimum.

*Proof.* As pointed in the paragraph below equation (B.3), when we restrict our attention to a particular row of $B$ and fix the rest of the rows the function $G_{\alpha,\beta,\mu}$ reduces to the function $h(\cdot)$ in equation (B.3) so that we can apply lemmas in Section B.1.

Concretely, fix an index $s \in [d]$ and let $x = b_s$. For all $i \in [d]$, let $\bar{\alpha}_i = \alpha_i \sum_{j \neq s}(b_j^\top e_i)^2$, and $\bar{\beta}_i = \mu\beta_i$. Then we have that

$$G_{\alpha,\beta,\mu}(B) = \sum_{i=1}^d \bar{\alpha}_i x_i^2 - \sum_i \bar{\beta}_i x_i^4 + \lambda\big(\|x\|^2 - 1\big)^2 \tag{B.11}$$

We view the function above as $h(x)$. Now we apply Lemma B.2 (by replacing $\alpha, \beta$ in Lemma B.2 by $\bar{\alpha}, \bar{\beta}$). The assumption that $\lambda_{\min}(\nabla^2 g_{\alpha,\beta,\mu}(\bar{B})) \geq -\tau$ implies that $\lambda_{\min}(\nabla^2 h(x)) \geq -\tau$ since $\nabla^2 h(x)$ is a submatrix of $\nabla^2 g(B)$. Moreover, $\|\nabla h(x)\| \leq \|\nabla G(B)\| \leq \varepsilon \leq \sqrt{\tau^3/(\mu\beta_{\min})}$

Hence by Lemma B.2, we have that the second largest entry of $|b_s|$ satisfies

$$\forall s, |b_s|_{\text{2nd}} \leq \delta. \tag{B.12}$$

where $\delta \triangleq \left(\frac{\tau}{\mu\beta_{\min}}\right)^{1/2}$ for the ease of notation. We can check that $\delta \leq \frac{1}{4\sqrt{\kappa_\alpha d}}$ by the assumption. Therefore, we have essentially shown that each row of $B$ has only one single large entry, since the second largest entry is at most $\delta$.

Next we show that each row of $B$ has largest entries on distinct columns. For each row $j \in [d]$, let $i_j = \arg\max_i |e_i^\top b_j|$ be the index of the largest entry of $b_j$. We will show that $i_1, \ldots, i_d$ are distinct.

For the sake of contradiction, suppose they are not distinct, that is, there are two distinct rows $s, t$ that have the same largest entries on column $l$, that is, we assume that $i_s = i_t = l$. This implies that $\{i_1, \ldots, i_d\} \neq [d]$ and let $k \in [d]$ be the index such that $k \notin \{i_1, \ldots, i_d\}$. We note that by the assumption $\delta = \left(\frac{\tau}{\mu\beta_{\min}}\right)^{1/2} \leq \frac{1}{4\sqrt{\kappa_\alpha d}} \leq \frac{1}{4\sqrt{d}}$. We first bound from above $\bar{\alpha}_k$

$$\bar{\alpha}_k = \alpha_k \sum_{j \neq s}(b_j^\top e_k)^2 \leq \alpha_k d\delta^2 \leq \frac{1}{16}\alpha_{\min}. \qquad \text{(by } \delta \leq \frac{1}{4\sqrt{\kappa_\alpha d}}\text{)}$$

Assume in addition without loss of generality that $|b_s^\top e_l| \leq |b_t^\top e_l|$. Let

$$z_l \triangleq \sum_{j \neq s} (b_j^\top e_l)^2 \tag{B.13}$$

be the sum of squares of the entries on the column $l$ without entry $b_j^\top e_l$, and that $\bar{\alpha}_l = \alpha_l z_l$. We first prove that $z_l \geq 1/3$.

For the sake of contradiction, assume $z_l < 1/3$. Then we have that $\bar{\alpha}_l = \alpha_l z \leq \frac{1}{3}\alpha_l$. This implies that $\lambda \geq 4\max\{\bar{\alpha}_l, \tau\}$, and since $l$ is the index of the largest column of $b_s$ we can invoke Lemma B.3 and conclude that $\|b_s\|^2 \geq 1/2$. This further implies that

$$(b_s^\top e_l)^2 \geq \|b_s\|^2 - d\,|b_s|_{\text{2nd}} \geq 1/2 - d\delta^2 \geq 1/3 \qquad \text{(by } \delta \leq 1/(4\sqrt{d}))$$

Since we have assumed that $|b_s^\top e_l| \leq |b_t^\top e_l|$. Then we obtain that

$$z_l \geq |b_t^\top e_l|^2 \geq |b_s^\top e_l|^2 \geq 1/3 \,,$$

which contradicts the assumption. Therefore, we conclude that $z_l \geq 1/3$. Then we are ready to bound $\bar{\alpha}_l$ from below:

$$\bar{\alpha}_l = \alpha_l z_l \geq \frac{1}{3}\alpha_l \,.$$

The display above and Equation (B.13) implies that

$$\bar{\alpha}_l - \bar{\alpha}_k \geq \frac{1}{4}\alpha_{\min}. \tag{B.14}$$

Note that $l$ is the largest entry in absolute value in the vector $b_s$. We will apply Lemma B.4. We fix every row of $B$ except $b_s$ and consider the objective as a function of $b_s$ only. Again let $\bar{\alpha}_i = \alpha_i \sum_{j \neq s}(b_j^\top e_i)^2$, and $\bar{\beta}_i = \mu\beta_i$ and we have the equation (B.11). (Note that now $\bar{\alpha}$ depends on the choice of $s$ which we fixed.) Lemma B.4 gives us that

$$\bar{\alpha}_l \leq \bar{\alpha}_k + 2\varepsilon + 2\tau + 4\bar{\beta}_\ell.$$

Since $\varepsilon \leq \frac{1}{50}\alpha_{\min}$, $\tau \leq \frac{1}{50}\alpha_{\min}$ and $\bar{\beta}_l = \mu\beta_l \leq \frac{1}{50}\alpha_{\min}$, we obtain that

$$\bar{\alpha}_l \leq \bar{\alpha}_k + \frac{1}{5}\alpha_{\min} \tag{B.15}$$

which contradicts equation (B.14). Thus we have established that $i_1, \ldots, i_d$ are distinct.

Finally, let $Q$ be the matrix that only contain the largest entries (in absolute value) of each columns of $B$. Since $i_1, \ldots, i_d$ are distinct, we have that $Q$ contains exactly one entry per row and per column. Therefore $Q$ can be written as $DP$ where $P$ is a permutation matrix and $D$ is a diagonal matrix. Moreover, we have that $\|b_s\|_\infty^2 \geq \|b_s\|^2 - d\,|b_s|_{\text{2nd}}^2 \geq 1/4$ and $\|b_s\|^2 \leq 2$. Therefore, the largest entry of each row has absolute value between $1/4$ and $2$. Therefore $|D|_{ii} \in [1/4, 2]$. Let $E = B - PD$. Then we have that $|E|_\infty \leq \max_s |b_s|_{\text{2nd}} \leq \delta$, which completes the proof.

$\square$

Applying Lemma B.5, we can further strengthen Proposition B.6 with better error bounds and better control of the largest entries of each column.

**Proposition B.7** (Strengthen of Proposition B.6). *In the setting of Proposition B.6. Suppose in addition that $\tau$ satisfies $\tau \leq c\mu\beta_{\min}^2/\beta_{\max}$. Then, the matrix $B$ can be written as*

$$B = DP + E \,,$$

*where $P$ is a permutation matrix, $D$ is diagonal such that*

$$\forall i \in [d], \quad \frac{1}{1 - \frac{\mu\beta_i}{\lambda}}\left(1 - \frac{18d\varepsilon^2}{\beta_{\min}^2} - \frac{\varepsilon}{2\lambda}\right) \leq |D_{ii}|^2 \leq \frac{1}{1 - \frac{\mu\beta_i}{\lambda}}\left(1 + \frac{\varepsilon}{2\lambda}\right)$$

*and*

$$|E|_\infty \leq \frac{3\varepsilon}{\beta_{\min}}.$$

*Proof.* By Proposition B.6, we know that $|E|_\infty \leq \delta = \left(\frac{\tau}{\mu\beta_{\min}}\right)^{1/2}$. Now we use Lemma B.5 to strength the error bound.

As we have done in the proof of Proposition B.6, we again fix an arbitrary $s \in [d]$ and all the rows except $b_s$ and view $G_{\alpha,\beta,\mu}$ as a function of $b_s$. For all $i \in [d]$, let $\bar{\alpha}_i = \alpha_i \sum_{j \neq s}(b_j^\top e_i)^2$, and $\bar{\beta}_i = \mu\beta_i$ and view $G_{\alpha,\beta,\mu}$ as a function of the form $h(x)$ with $\alpha, \beta$ replaced by $\bar{\alpha}, \bar{\beta}$, namely,

$$h(x) = \sum_k \bar{\alpha}_k x_k^2 - \sum_k \bar{\beta}_k x_k^4 + \lambda\big(\|x\|^2 - 1\big)^2 + \text{const}$$

We will verify the condition of Lemma B.5. Let $i$ be the index of the largest entry in absolute value of the vector $b_s$. Since we have shown that the largest entry in each row sits on different columns, and the second largest entry is always less than $\delta$, we have that,

$$\bar{\alpha}_i = \alpha_i \sum_{j \neq s}(b_j^\top e_i)^2 \leq \alpha_i d\delta^2 \leq \frac{1}{16}\alpha_{\min}. \qquad \text{(by } \delta \leq \frac{1}{4\sqrt{\kappa_\alpha d}})$$

For any $k \neq i$, we know that the column $k$ contains some entry $(k, j_k)$ which is the largest entry of some row, and we also have that $j_k \neq s$ since the largest entry of row $s$ is on column $i$. Therefore, we have that

$$\bar{\alpha}_k = \alpha_k \sum_{j \neq s}(b_j^\top e_k)^2 \geq \alpha_l(b_{j_k}^\top e_k)^2 \geq \alpha_l(\|b_k\|^2 - d\delta^2)$$

$$\geq \frac{1}{3}\alpha_l \qquad \text{(by } \delta \leq 1/(4\sqrt{d}))$$

Therefore, $\bar{\alpha}_k \geq \bar{\alpha}_i$ for any $k \neq i$ and thus $i = \arg\min_k |\bar{\alpha}_k|$. By the fact that $|E|_\infty \leq \delta$, we have that $\|x_{-i}\|_\infty \leq \delta \leq 0.1\min\{1/\sqrt{d}, \sqrt{\beta_{\min}/(\beta_{\max})}\}$. Now we are ready to apply Lemma B.5 and obtain that $|b_s|_{2\text{nd}} \leq \frac{3\varepsilon}{\beta_{\min}}$. Applying the argument for every row $s$ gives $|E|_\infty \leq \frac{3\varepsilon}{\beta_{\min}}$.

Finally, we give the bound for the entires in $D$. Let $v$ be a short hand for $\nabla h(b_s)$ which is equal to the $s$-th column of $\nabla G(B)$. Since $B$ is an $\varepsilon$-approximate stationary point, then we have that $\|v\| \leq \varepsilon$ and by straightforward calculation of the gradient, we have

$$v_i = 2\bar{\alpha}_i x_i - 4\mu\beta_i x_i^3 + 4\lambda(\sum_{j=1}^d x_j^2 - 1)x_i.$$

Since $x_i \neq 0$, dividing by $x_i$ gives,

$$0 = 2\bar{\alpha}_i - 4\mu\beta_i x_i^2 + 4\lambda(\sum_{j=1}^d x_j^2 - 1) - \frac{v_i}{x_i}$$

$$= (4\lambda - 4\mu\beta_i)x_i^2 + 4\lambda\sum_{j \neq i} x_j^2 + 2\bar{\alpha}_i - 4\lambda - \frac{v_i}{x_i}$$

Rearranging the equation above gives,

$$x_i^2 = \frac{1}{4\lambda - 4\mu\beta_i}\left(4\lambda - 2\bar{\alpha}_i - 4\lambda\sum_{j \neq i} x_j^2 - \frac{v_i}{x_i}\right)$$

$$= \frac{1}{1 - \frac{\mu\beta_i}{\lambda}}\left(1 - \frac{\bar{\alpha}_i}{2\lambda} - \sum_{j \neq i} x_j^2 - \frac{v_i}{4\lambda x_i}\right)$$

To upper bound $x_i^2$, we note that $|v_i| < \varepsilon$, $\bar{\alpha}_i > 0$, and $\sum_{j \neq i} x_j^2 > 0$, so

$$x_i^2 \leq \frac{1}{\left(1 - \frac{\mu\beta_i}{\lambda}\right)}\left(1 + \frac{\varepsilon}{2\lambda}\right) \leq 1 + \frac{2\mu\beta_i + \varepsilon}{\lambda} \qquad \text{(since } \lambda \geq 4\mu\beta_i)$$

For the lower bound of $x_i^2$, we note that $|E|_\infty \le \delta = \frac{3\varepsilon}{\beta_{\min}}$ implies $\sum_{j\ne i} x_j^2 \le d\delta^2$. Moreover, we have proved that each rows has largest entry at different columns. Also note that the largest entry of row $b_s$ is on column $i$. Therefore, we have $\bar{\alpha}_i = \alpha_i \sum_{j\ne s}(b_j^T e_i)^2 \le \alpha_{\max} d\delta^2$. Using these two estimates and $\delta = \frac{3\varepsilon}{\beta_{\min}}$, we have

$$x_i^2 \ge \frac{1}{1 - \frac{\mu\beta_i}{\lambda}}(1 - (\frac{\alpha_{\max}}{2\lambda} + 1)d\delta^2 - \frac{\varepsilon}{2\lambda})$$

$$= \frac{1}{1 - \frac{\mu\beta_i}{\lambda}}\left(1 - \frac{18d\varepsilon^2}{\beta_{\min}^2} - \frac{\varepsilon}{2\lambda}\right)$$

$\square$

Finally we are ready to prove Theorem B.1 by applying Proposition B.6.

*Proof of Theorem B.1.* By setting $\varepsilon = 0, \tau = 0$ in Proposition B.6, we have that any local minimum $B$ satisfies that $B = DP$ where $P$ is a permutation matrix and $D$ is a diagonal and the precise diagonal entries of $D$. It can be verified that all these points have the same function value, so that they are all global minimizers.

Towards proving the second bullet, we note that a saddle point $B$ satisfies that $\nabla G(B) = 0$. We will prove that $\lambda_{\min}(\nabla^2 G(B)) \le -\tau_0$. For the sake of contradiction, suppose $\lambda_{\min}(\nabla^2 G(B)) \ge -\tau_0$. Then setting $\varepsilon = 0$ and $\tau = \tau_0$ in Propostion B.7, we have that $B = DP$ and $D_{ii} = \left\{\pm\sqrt{\frac{1}{1-\mu\beta_i/\lambda}}\right\}$, which by bullet 1 implies that $B$ is a local minimum. This contradicts the assumption that $B$ is a saddle point.

The 3rd bullet is a just a rephrasing of Proposition B.7. $\square$

## C   HANDLING NON-ORTHOGONAL WEIGHTS

In this section, we first show that when the weight vectors $\{b_i^\star\}'s$ are not orthonormal, the local optimum of a slight variant of $G(B)$ still allow us to recover $B^\star$. The main observation is that the set of local minima are preserved (in a certain sense) by linear transformation of the variables. We design an objective function $F(B)$ that is equivalent to $G(B)$ up to a linear transformation. This allows us to use Theorem 2.3 as a black box to characterize all the local minima of $F$.

We use $\lambda_{\max}(\cdot), \lambda_{\min}(\cdot)$ to denote the largest and smallest eigenvalues of a square matrix. Similarly, $\sigma_{\max}(\cdot)$ and $\sigma_{\min}(\cdot)$ are used to denote the largest and smallest singular values.

### C.1   LOCAL MINIMUM AFTER A LINEAR TRANSFORMATION

Given a function $f(y)$, we say function $g(\cdot)$ is a linear transformation of $f(\cdot)$ if there is a matrix $W$ such that $g(x) = f(Wx)$. If $W$ has full rank, the local minima of $f$ are closely related to the local minima of $g$.

We recall some standard notation in calculus first. We use $\nabla f(t)$ to denote the gradient of $f$ evaluated at $t$. For example, $\nabla f(Wx)$ is a shorthand for $\frac{\partial f(y)}{\partial y}|_{y=Wx}$, and similarly $\nabla^2 f(Wx)$ is $\frac{\partial^2 f(y)}{(\partial y)^2}|_{y=Wx}$.

The following theorem then connects the gradients and Hessians of $f(Wx)$ and $g(x)$. Essentially, it shows that the set of local minima and saddle points have a 1-1 mapping between $f$ and $g$, and the corresponding norms/eigenvalues only differ multiplicatively by quantities related to the spectrum of $W$.

**Theorem C.1.** *Let $W \in \mathbb{R}^{d\times m}(d \ge m)$ be a full rank matrix. Suppose $g : \mathbb{R}^m \to \mathbb{R}$ and $f : \mathbb{R}^d \to \mathbb{R}$ are twice-differentiable functions such that $g(x) = f(Wx)$ for any $x \in \mathbb{R}^m$. Then, for all $x \in \mathbb{R}^m$, the following three properties hold:*

   *1. $\sigma_{min}(W)\|\nabla f(Wx)\| \le \|\nabla g(x)\| \le \sigma_{\max}(W)\|\nabla f(Wx)\|$.*

2. *If $\lambda_{min}(\nabla^2 g(x)) < 0$, then*

$$\sigma_{\max}(W)^2 \lambda_{min}(\nabla^2 f(Wx)) \leq \lambda_{min}(\nabla^2 g(x)) \leq \sigma_{min}(W)^2 \lambda_{min}(\nabla^2 f(Wx)).$$

3. *The point $x$ satisfies the first and second order optimality condition for $g$ iff $y = Wx$ also satisfy the first and second order optimality condition for $f$.*

*Proof.* The proof follows from the relationship between the gradients of $g$ and the gradients of $f$. By basic calculus, we have

$$\nabla g(x) = \frac{\partial f(Wx)}{\partial x} = W^\top \frac{\partial f(y)}{\partial y}\Big|_{y=Wx} = W^\top \nabla f(Wx)$$

which immediately implies bullet 1. Similarly, we can compute the second order derivative:

$$\nabla^2 g(x) = W^\top [\nabla^2 f(Wx)]W.$$

To simplify notation, let $A = \nabla^2 f(Wx)$. Let $x = \arg\min_{\|x\|=1} x^\top W^\top AWx$, and $y = (Wx)/\|Wx\|$. Therefore

$$\lambda_{min}(A) \leq y^\top Ay \leq \lambda_{min}(W^\top AW)/\|Wx\|^2 \leq \lambda_{\min}(W^\top AW)/\|W\|^2.$$

On the other hand, let $y$ be the unit vector that minimizes $y^\top Ay$, we know $y$ is in column span of $W$ because $f$ is only defined on the row span, so there must exist a unit vector $x$ such that $Wx = \lambda y$ where $\lambda \geq \sigma_{min}(W)$. For this $x$ we have $\lambda_{min}(W^\top AW) \leq x^\top W^\top AWx = \lambda^2 \lambda_{min}(A) \leq \sigma_{min}^2(W)\lambda_{min}(A)$. This finishes the proof for 2.

Finally, notice that $W$ is full rank, so $\nabla g(x) = W^\top \nabla f(Wx) = 0$ iff $\nabla f(Wx) = 0$. Also, $\nabla^2 g(x) = W^\top [\nabla^2 f(Wx)]W \succeq 0$ iff $\nabla^2 f(Wx) \succeq 0$. □

## C.2 OBJECTIVE FOR NON-ORTHOGONAL WEIGHTS

Now we will design a new objective function that can be linearly transformed to the orthonormal case. The main idea is to view the rows of $B^\star$ as the new basis that we work on (which is not necessarily orthogonal). Note that this is already the case for the first two terms of the objective function $G(B)$, we change the objective function as follows: More concretely, we define

$$F_{\alpha,\mu,\lambda}(B) = 2\sqrt{6}\hat{\sigma} \cdot \sum_{i\in[d]} \alpha_i \sum_{j,k\in[d]} \langle b_i^\star, b_j\rangle^2 \langle b_i^\star, b_k\rangle^2$$

$$- \frac{\hat{\sigma}_4\mu}{\sqrt{6}} \sum_{i,j\in[d]} \alpha_i \langle b_i^\star, b_j\rangle^4 + \lambda \sum_{j=1}^m ((\sum_{i=1}^m \alpha_i\langle b_j, b_i^\star\rangle^2 - 1)^2 - 1)^2 .$$

Note that the only change in the objective is the regularizer for the norm of $b_j$. It is now replaced by $((\sum_{i=1}^m \alpha_i\langle b_j, b_i^\star\rangle^2 - 1)^2 - 1)^2$, which tries to ensure the "norm" of $b_j$ in the basis defined by row of $B^\star$ to be 1. The objective function that we will optimize corresponds to choosing $\alpha_i = a_i^\star$.

Similar as before, this function can be computed as expectations

$$F_{a^\star,\mu,\lambda}(B) = \mathbb{E}\left[y \cdot \sum_{j,k\in[d],j\neq k} \phi(b_j, b_k, x)\right] - \mu\,\mathbb{E}\left[y \cdot \sum_{j\in[d]} \varphi(b_j, x)\right]$$

$$+ \lambda\mathbb{E}_{(x,y),(x',y')}[\sum_{i=1}^m y \cdot \phi_2(b_i, x) \cdot y' \cdot \phi_2(b_i, x')], \tag{C.1}$$

where $(x', y')$ is an independent sample, and $\phi_2(v, x) = (v^\top x)^2 - \|v\|^2$.

Intuitively, if we can find a linear transformation that makes $\{b_i^\star\}$'s orthonormal, that will reduce the problem to the orthonormal case. This is in fact the whitening matrix:

Let $M = \sum_{i=1}^{m} a_i^{\star} b_i^{\star} (b_i^{\star})^{\top}$ be the weighted covariance matrix of $b_i^{\star}$'s. Suppose the SVD of $M$ is $UDU^{\top}$ and let $W = UD^{-1/2}$. We apply the transformation $W^{\top}$ to the vectors $\sqrt{a_i^{\star}} b_i$'s and obtain that $o_i = W^{\top} \sqrt{a_i^{\star}} b_i^{\star}$. We can verify that $o_i$'s are orthogonal vectors because

$$\sum_{i \in [m]} o_i o_i^{\top} = W^{\top} M W = \mathrm{Id} \tag{C.2}$$

For notational convenience, let's extend the definition of the $G(\mathcal{B})$ in equation by using the putting the relevant information in the subscript

$$G_{\alpha, \beta, \lambda, o}(\mathcal{B}) = \sqrt{6} \hat{\sigma} \cdot \sum_{i \in [d]} a_i^{\star} \sum_{j, k \in [d], j \neq k} \langle o_i, \bar{b}_j \rangle^2 \langle o_i, \bar{b}_k \rangle^2 - \frac{\hat{\sigma}_4 \mu}{\sqrt{6}} \sum_{i, j \in [d]} a_i^{\star} \langle o_i, \bar{b}_j \rangle^4 \,.$$

$$+ \lambda \sum_{i=1}^{m} (\|\bar{b}_i\|^2 - 1)^2$$

(That is, the index $o$ denotes the ground-truth solution with respect to which $G$ is defined.)

The next Theorem shows that we can rotate the objective function $F$ properly so that it matches the objective $G$ with a ground-truth vector $o_i$'s.

**Theorem C.2.** *Let $W$ be defined as above, and let $1/a^{\star}$ be the vector whose $i$-th entry is $1/a_i^{\star}$. Then, we have that*

$$G_{1/a^{\star}, \mu, \lambda, o_i}(\mathcal{B}) = F_{a^{\star}, \mu, \lambda}(\mathcal{B} W^{\top})).$$

*Note this can be interpreted as a linear transformation as in vector format $\mathcal{B} W^{\top}$ is equal to $\mathcal{B} \cdot (W^{\top} \otimes \mathrm{Id}_{d \times d})$.*

*Proof.* The equality can be obtained by straightforward calculation. We note that since $\mathcal{B} = \begin{bmatrix} \bar{b}_1^{\top} \\ \vdots \\ \bar{b}_m^{\top} \end{bmatrix}$,

the rows of $\mathcal{B} \cdot (W^{\top} \otimes \mathrm{Id}_{d \times d})$ are $W \bar{b}_1, \dots, W \bar{b}_m$.

Therefore, we have that

$$F_{a^{\star}, \mu, \lambda}(\mathcal{B} \cdot (W^{\top} \otimes \mathrm{Id}_{d \times d})) \tag{C.3}$$

$$= 2\sqrt{6} \hat{\sigma} \cdot \sum_{i \in [d]} a_i^{\star} \sum_{j \neq k \in [d]} \langle b_i^{\star}, W \bar{b}_j \rangle^2 \langle b_i^{\star}, W \bar{b}_k \rangle^2$$

$$- \frac{\hat{\sigma}_4 \mu}{\sqrt{6}} \sum_{i, j \in [d]} a_i^{\star} \langle b_i^{\star}, W \bar{b}_j \rangle^4 + \lambda \sum_{j=1}^{m} (\sum_{i=1}^{m} a_i^{\star} \langle W \bar{b}_j, b_i^{\star} \rangle^2 - 1)^2 \,.$$

$$= 2\sqrt{6} \hat{\sigma} \cdot \sum_{i \in [d]} \frac{1}{a_i^{\star}} \sum_{j, k \in [d]} \langle \sqrt{a_i^{\star}} W^{\top} b_i^{\star}, \bar{b}_j \rangle^2 \langle \sqrt{a_i^{\star}} W^{\top} b_i^{\star}, \bar{b}_k \rangle^2$$

$$- \frac{\hat{\sigma}_4 \mu}{\sqrt{6}} \sum_{i, j \in [d]} \frac{1}{a_i^{\star}} \langle \sqrt{a_i^{\star}} W^{\top} b_i^{\star}, \bar{b}_j \rangle^4 + \lambda \sum_{j=1}^{m} (\sum_{i=1}^{m} \langle \bar{b}_j, \sqrt{a_i^{\star}} W^{\top} b_i^{\star} \rangle^2 - 1)^2 \,.$$

$$= 2\sqrt{6} \hat{\sigma} \cdot \sum_{i \in [d]} \frac{1}{a_i^{\star}} \sum_{j, k \in [d]} \langle o_i, \bar{b}_j \rangle^2 \langle o_i, \bar{b}_k \rangle^2$$

$$- \frac{\hat{\sigma}_4 \mu}{\sqrt{6}} \sum_{i, j \in [d]} \frac{1}{a_i^{\star}} \langle o_i, \bar{b}_j \rangle^4 + \lambda \sum_{j=1}^{m} (\sum_{i=1}^{m} \langle \bar{b}_j, o_i \rangle^2 - 1)^2 \,. \qquad \text{(by the definition of } o_i\text{'s)}$$

$$\square$$

From Theorem 2.3 we can immediately get the following Corollary (note that the only difference is that the coefficients now are $1/a_i^{\star}$ instead of $a_i^{\star}$). Recall $a_{max}^{\star} = \max_i a_i^{\star}$ and $a_{min}^{\star} = \min_i a_{min}^{\star}$, we have

**Corollary C.3.** *Let $\kappa_a = a^\star_{max}/a^\star_{min}$. Let $c$ be a sufficiently small universal constant (e.g. $c = 0.01$ suffices). Assume $\mu \leq c/\kappa_a$ and $\lambda \geq (ca^\star_{min})^{-1}$. The function $G_{1/a^\star,\mu,\lambda,o_i}(\cdot)$ defined as in Theorem C.2 satisfies that*

1. *A matrix $\mathcal{B}$ is a local minimum of $G$ if and only if $\mathcal{B}$ can be written as $\mathcal{B} = PDO$ where $O$ is a matrix whose rows are $o_i$'s, $P$ is a permutation matrix and $D$ is a diagonal matrix with $D_{ii} \in \{\pm 1 \pm O(\mu/\lambda a^\star_{min})\}$.*

2. *Any saddle point $\mathcal{B}$ has a strictly negative curvature in the sense that $\lambda_{\min}(\nabla^2 G(\mathcal{B})) \geq -\tau_0$ where $\tau_0 = c \min\{\mu/(\kappa_a a^\star_{max}d), \lambda\}$*

3. *Suppose $\mathcal{B}$ is an approximate local minimum in the sense that $\mathcal{B}$ satisfies*

$$\|\nabla g(\mathcal{B})\| \leq \varepsilon \text{ and } \lambda_{\min}(\nabla^2 g(\mathcal{B})) \geq -\tau_0$$

   *Then $\mathcal{B}$ can be written as $\mathcal{B} = PDO + E$ where $P$ is a permutation matrix, $D$ is a diagonal matrix and $|E|_\infty \leq O(\varepsilon a^\star_{max}/\hat{\sigma}_4)$.*

Finally, we can combine the theorem above and Theorem B.1 to give a guarantee for optimizing $F$. Let $\Gamma$ be a diagonal matrix with $\Gamma_{ii} = \sqrt{a^\star_i}$. Let $M = B^{\star \top}\Gamma^2 B^\star$ and $\kappa(M) = \|M\|/\sigma_{min}(M)$.

**Theorem C.4.** *Let $c$ be a sufficiently small universal constant (e.g. $c = 0.01$ suffices). Let $\kappa_a = a^\star_{max}/a^\star_{min}$. Assume $\mu \leq c/\kappa_a$ and $\lambda \geq 1/(c \cdot a^\star_{\min})$. The function $F(\cdot)$ defined as in Theorem C.2 satisfies that*

1. *A matrix $B$ is a local minimum of $F$ if and only if $B$ satisfy $B^{-\top} = PD\Gamma B^\star$ where $P$ is a permutation matrix, $\Gamma$ is a diagonal matrix with $\Gamma_{ii} = \sqrt{a^\star_i}$, and $D$ is a diagonal matrix with $D_{ii} \in \{\pm 1 \pm O(\mu/\lambda a^\star_{min})\}$. Furthermore, this means that all local minima of $F$ are also global.*

2. *Any saddle point $B$ has a strictly negative curvature in the sense that $\lambda_{\min}(\nabla^2 F(B)) \geq -\tau_0$ where $\tau_0 = c \min\{\mu/(\kappa_a d a^\star_{\max}), \lambda\}\sigma_{min}(M)$.*

3. *Suppose $B$ is an approximate local minimum in the sense that $B$ satisfies*

$$\|\nabla F(B)\| \leq \varepsilon \text{ and } \lambda_{\min}(\nabla^2 F(B)) \geq -\tau_0$$

   *Then $B$ can be written as $B^{-\top} = PD\Gamma B^\star + E$ where $\Gamma, D, P$ are as in 1, the error term $\|E\| \leq O(\varepsilon a^\star_{\max}\sqrt{md} \cdot \kappa(M)^{1/2}/\hat{\sigma}_4)$ (when $\varepsilon a^\star_{\max}\sqrt{md} \cdot \kappa(M)^{1/2}/\hat{\sigma}_4 < c$).*

*Proof.* Note that we can immediately apply Theorem 2.3 to $G_{1/a^\star,\mu,\lambda,o_i}(B)$ to characterize all its local minima. See Corollary C.3.

Next we will transform the properties for local minima of $G$ (stated in Corollary C.3) to $F$ using Theorem C.1. First we note that the transformation matrix $W$ and $M$ are closely related:

$$WW^\top = M, \sigma_{min}(W)^2 = 1/\|M\|, \|W\|^2 = 1/\sigma_{min}(M). \tag{C.4}$$

This is because according to the definition of $W$, the SVD of $M$ is $M = UDU^\top$ and $W = UD^{-1/2}$, so $WW^\top = UD^{-1}U^\top = M^{-1}$. The claims of the singular values follow immediately from the SVD of $M$ and $W$.

As a result, all local minimum of $F$ are of the form $\mathcal{B}W^\top$ where $\mathcal{B}$ is a local minimum of $G$. For $B = \mathcal{B}W^\top$, the gradient and Hessian of $F(B)$ and $G(\mathcal{B})$ are also related by Theorem C.1.

Let us first prove 1. By Corollary C.3, we know every local minimum of $G$ is of the form $\mathcal{B} = PDO$. According to the definition of $O$ in Theorem C.2, we know each row vector $o_i$ is equal to $W^\top(a^\star_i)^{1/2}b^\star_i$, therefore $O = \Gamma B^\star W$. As a result, all local minima of $G$ are of the form $\mathcal{B} = PD\Gamma B^\star W$. By Theorem C.1 and Theorem C.2, we know all local minima of $F$ must be of the form $B = \mathcal{B}W^\top = PD\Gamma B^\star WW^\top = PD\Gamma B^\star M^{-1}$.

Now we try to compute $B^{-\top}$. To do that observe that $[\Gamma B^\star]M^{-1}[\Gamma B^\star]^\top = I$. Therefore $[\Gamma B^\star M^{-1}]^{-\top} = \Gamma B^\star$, and for any local minimum $B$, we have

$$B^{-\top} = (PD\Gamma B^\star M^{-1})^{-\top} = P^{-\top}D^{-\top}(\Gamma B^\star M^{-1})^{-\top}$$

$$= P^{-\top}D^{-\top}\Gamma B^{\star}.$$

Note that $P^{\top}$ is still a permutation matrix, and $D^{-\top}$ is still a matrix whose diagonal entries are $\{\pm 1 \pm O(\mu/\lambda a^{\star}_{\min})\}$, so this is exactly the form we stated in 1. More concretely, the rows of $B^{-\top}$ are permutations of $\sqrt{a^{\star}_i}b^{\star}_i$.

For bullet 2, it follows immediately from Property 2 in Theorem C.1. Note that by property 2,

$$\lambda_{min}(\nabla^2 F(\mathcal{B}W^{\top})) \leq \frac{\lambda_{min}(\nabla^2 G(\mathcal{B}))}{\|W\|^2} = \lambda_{min}(\nabla^2 G(\mathcal{B}))\sigma_{min}(M).$$

Finally we will prove 3. Let $\mathcal{B} = BW^{-\top}$, so that $G(\mathcal{B}) = F(B)$. We will prove properties of $B$ using the properties of $\mathcal{B}$ from Corollary C.3.

First we observe that by Theorem C.1,

$$\lambda_{min}(\nabla^2 G(\mathcal{B})) \geq \|W\|^2 \lambda_{min}(\nabla^2 F(B)) \geq -c\min\{\mu/(\kappa_a da^{\star}_{\max}, \lambda\}.$$

Therefore the second order condition for Claim 3 in Corollary C.3 is satisfied. Now when $\|\nabla F(B)\| \leq \varepsilon$, we have $\|\nabla G(\mathcal{B})\| \leq \varepsilon\|W\| = \varepsilon/\sigma_{min}(M)^{1/2}$. By Corollary C.3, we know $\mathcal{B}$ can be expressed as $PDO + E'$ where $D$ is the diagonal matrix, $P$ is a permutation matrix and $|E'|_{\infty} \leq O(\varepsilon a^{\star}_{\max}/(\hat{\sigma}_4 \sigma_{min}(M)^{1/2}))$. We will apply perturbation Theorem C.9 for matrix inversion. Since $\sigma_{min}(PDO) \geq 1/2$, we know when $\|E'\| \leq 1/4$,

$$\|(PDO + E')^{-1} - (PDO)^{-1}\| \leq 8\sqrt{2}\|E'\|.$$

Here $\|E\|$ is bounded by $\|E\|_F \leq \sqrt{md}|E'|_{\infty} \leq O(\varepsilon a^{\star}_{\max}\sqrt{md}/(\hat{\sigma}_4 \sigma_{min}(M)^{1/2}))$, which is smaller than $1/4$ when $\varepsilon$ is small enough.

The corresponding point in $F$ is $B = \mathcal{B}W^{\top}$, and in 1 we have already proved $(PDOW^{\top})^{-\top}$ is of the form we want, therefore we can define $E = B^{-\top} - (PDOW^{\top})^{-\top} = (\mathcal{B} - PDO)^{-\top}W^{-1}$, and

$$\|E\| = \|W^{-1}\|\|(PDO + E')^{-1} - (PDO)^{-1}\| = O(\varepsilon a^{\star}_{\max}\sqrt{md}\cdot\kappa(M)^{1/2}/\hat{\sigma}_4).$$

This finishes the proof. $\qquad\qquad\qquad\qquad\qquad\qquad\qquad\qquad\qquad\qquad\qquad\qquad\qquad\square$

## C.3 HANDLE UNDERCOMPLETE CASE

The objective function $F$ can handle the case when the weights $b^{\star}_i$'s are not orthogonal, but still requires the number of components $m$ to be equal to the number of dimensions $d$. In this section we show how to use similar ideas for the case when the number of components is smaller than the dimension ($m < d$).

Note that all the terms in $F(B)$ only depends on the inner-products $\langle b_j, b^{\star}_i\rangle$. Let $\mathcal{S}$ be the span of $\{b^{\star}_i\}$'s and $P_{\mathcal{S}}$ be the projection matrix to this subspace, it is easy to see that $F(B)$ satisfies

$$F(B) = F(BP_{\mathcal{S}}).$$

That is, the previous objective function only depends on the projection of $B$ in the space $\mathcal{S}$. Using similar argument as Theorem C.4, it is not hard to show the only local optimum in $\mathcal{S}$ satisfies the same conditions, and allow us to recover $B^{\star}$. However, without modifying the objective, the local optimum of $F(B)$ can have arbitrary components in the orthogonal subspace $\mathcal{S}^{\perp}$.

In order to prevent the components from $\mathcal{S}^{\perp}$, we add an additional $\ell_2$ regularizer: define $\mathcal{F}_{\alpha,\mu,\lambda,\delta}$ as follows:

$$\mathcal{F}_{\alpha,\mu,\lambda,\delta}(B) = F_{\alpha,\mu,\lambda}(B) + \frac{\delta}{2}\|B\|^2_F \qquad\qquad\qquad\text{(C.5)}$$

Intuitively, since the first term $F_{\alpha,\mu,\lambda}(B)$ only cares about the projection $BP_{\mathcal{S}}$, minimizing $\|B\|^2_F$ will remove the components in the orthogonal subspace of $\mathcal{S}$. We will choose $\delta$ carefully to make

sure that the additional term does not change the local optima of $F_{\alpha,\mu,\lambda}(B)$ by too much, while still ensuring a small projection on $\mathcal{S}^{\perp}$.

In this case we will consider pseudo-inverse instead of inverse. In particular, for a $m \times d$ matrix $B$, define its pseudo-inverse $B^{\dagger}$ to be the matrix such that $BB^{\dagger} = \text{Id}_{m \times m}$ and $B^{\dagger}B$ is the projection to the row span of $B$.

Let $M = \sum_{i=1}^{m} a_i^{\star} b_i^{\star}(b_i^{\star})^{\top}$, $\kappa(M) = \|M\|/\sigma_m(M)$.

**Theorem C.5.** *For any desired accuracy $\varepsilon_0$, we can choose parameters $\varepsilon, \delta, \tau_0, \mu, \lambda$, such that for the objective function $\mathcal{F}_{a^{\star},\mu,\lambda,\delta}(B)$, for any $B$ such that*

$$\|\nabla \mathcal{F}(B)\| \leq \varepsilon, \quad \nabla^2 \mathcal{F}(B) \geq -\tau_0/2,$$

*we have $[B^{\dagger}]^{\top} = B^{\star}D\Gamma P + E$ where $\Gamma$ is a diagonal matrix with entries $\sqrt{a_i^{\star}}$, $D$ is a diagonal matrix with entries close to 1, $P$ is a permutation matrix and $\|E\| \leq \varepsilon_0$.*

*To choose the parameters, let $c$ be a sufficiently small universal constant (e.g. $c = 0.01$ suffices). Assume $\mu \leq c/\kappa^{\star}$ and $\lambda \geq 1/(c \cdot a_{\min}^{\star})$. Let $\tau_0 = c \min\{\mu/(\kappa d a_{\max}^{\star}), \lambda\}\sigma_{min}(M)$. Let $\delta \leq \min\{\frac{c\hat{\sigma}_4 \varepsilon_0}{a_{max}^{\star} \cdot m\sqrt{d}\kappa^{1/2}(M)}, \tau_0/2\}$, and $\varepsilon = \min\{\lambda\sigma_{min}(M)^{1/2}, c\delta/\sqrt{\|M\|}, c\varepsilon_0\delta\sigma_{min}(M)\}$.*

We first show that if the gradient is small, then the point cannot have a large component in $\mathcal{S}^{\perp}$.

**Lemma C.6.** *If $\|\nabla \mathcal{F}_{a^{\star},\mu,\lambda,\delta}(B)\| \leq \varepsilon$, then $\|P_{\mathcal{S}^{\perp}}B\|_F \leq \varepsilon/\delta$.*

*Proof.* Since $F_{a^{\star},\mu,\lambda}(B)$ only depends $BP_{\mathcal{S}}$, we know $\nabla F_{a^{\star},\mu,\lambda}(B)P_{\mathcal{S}^{\perp}} = 0$. Therefore $\varepsilon \geq \|\nabla \mathcal{F}_{a^{\star},\mu,\lambda,\delta}(B)P_{\mathcal{S}^{\perp}}\|_F = \|(\delta B)P_{\mathcal{S}^{\perp}}\|_F = \delta\|P_{\mathcal{S}^{\perp}}B\|_F$, and we have $\|BP_{\mathcal{S}^{\perp}}\|_F \leq \varepsilon/\delta$ as desired. $\square$

Next we show that if the gradient of $\mathcal{F}_{a^{\star},\mu,\lambda,\delta}(B)$ is small, and $\delta$ is also small, then the gradient of $F_{a^{\star},\mu,\lambda}(B)$ can be bounded.

**Lemma C.7.** *In the setting of Theorem C.5, if $\|\nabla \mathcal{F}_{a^{\star},\mu,\lambda,\delta}(B)\| \leq \varepsilon \leq \lambda\sigma_{min}(M)^{1/2}$, then we have*

$$\|\nabla F_{a^{\star},\mu,\lambda}(B)\| \leq \varepsilon + \delta\sqrt{2m/\sigma_{min}(M)}.$$

Towards proving Lemma C.7, we first bound the norm of $B$ by the following claim:

*Claim* C.8. If $\|\nabla \mathcal{F}_{a^{\star},\mu,\lambda,\delta}(B)\| \leq \lambda\sigma_{min}(M)^{1/2}$, then each row $b_i$ must satisfy $b_i^{\top}Mb_i \leq 2$.

*Proof.* We prove by contradiction. Assume towards contradiction that there is a column $b_i$ such that $b_i^{\top}Mb_i \geq 2$. We consider the quantity,

$$\langle \frac{\partial \mathcal{F}_{a^{\star},\mu,\lambda,\delta}}{\partial b_i}(B), b_i \rangle.$$

Note that $\mathcal{F}_{a^{\star},\mu,\lambda,\delta}$ has 4 terms: (1) $2\sqrt{6}\hat{\sigma} \cdot \sum_{i \in [d]} \alpha_i \sum_{j,k \in [d]} \langle b_i^{\star}, b_j \rangle^2 \langle b_i^{\star}, b_k \rangle^2$, (2) $-\frac{\hat{\sigma}_4 \mu}{\sqrt{6}} \sum_{i,j \in [d]} \alpha_i \langle b_i^{\star}, b_j \rangle^4$, (3) $\lambda \sum_{j=1}^{m}((\sum_{i=1}^{m} \alpha_i \langle b_j, b_i^{\star} \rangle^2 - 1)^2 - 1)^2$, (4) $\frac{\delta}{2}\|B\|_F^2$.

Among these 4 terms, the first, third and forth terms all contribute positively to this inner-product (because when $b_i$ is moved to $(1 - \varepsilon)b_i$ all those terms clearly decrease). Term 2 $-\frac{\hat{\sigma}_4 \mu}{\sqrt{6}} \sum_{i,j \in [m]} a_i^{\star}\langle b_i^{\star}, b_j \rangle^4$ contribute negatively. Therefore we can ignore terms 1 and 4:

$$\langle \frac{\partial \mathcal{F}_{a^{\star},\mu,\lambda,\delta}}{\partial b_i}(B), b_i \rangle \geq \langle \frac{\partial}{\partial b_i}[-\frac{\hat{\sigma}_4 \mu}{\sqrt{6}} \sum_{i \in [m]} a_i^{\star}\langle b_i^{\star}, b_j \rangle^4 + \lambda(\sum_{i \in [m]} a_i^{\star}\langle b_i^{\star}, b_j \rangle^2 - 1)^2], b_i \rangle.$$

Let $b_i^{\top}Mb_i = C \geq 2$, we know $\sum_{i \in [m]} a_i^{\star}\langle b_i^{\star}, b_j \rangle^4 \leq \frac{1}{a_{min}^{\star}} \sum_{i \in [m]} (a_i^{\star})^2\langle b_i^{\star}, b_j \rangle^4 \leq C^2/a_{min}^{\star}$. Therefore,

$$\langle \frac{\partial}{\partial b_i}[-\frac{\hat{\sigma}_4 \mu}{\sqrt{6}} \sum_{i \in [m]} a_i^{\star}\langle b_i^{\star}, b_j \rangle^4], b_i \rangle = -\frac{4\hat{\sigma}_4 \mu}{\sqrt{6}} \sum_{i \in [m]} a_i^{\star}\langle b_i^{\star}, b_j \rangle^4 \geq -\frac{4\hat{\sigma}_4 \mu}{\sqrt{6}} \cdot \frac{C^2}{a_{min}^{\star}}$$

On the other hand,

$$\langle \frac{\partial}{\partial b_i} [\lambda (\sum_{i \in [m]} a_i^\star \langle b_i^\star, b_j \rangle^2 - 1)^2, b_i \rangle = 4\lambda (b_i^\top M b_i - 1)(b_i^\top M b_i) = 4\lambda C(C-1).$$

By the choice of $\lambda, \mu$, we can see that the negative term is negligible, and we know

$$\langle \frac{\partial \mathcal{F}_{a^\star, \mu, \lambda, \delta}}{\partial b_i}(B), b_i \rangle \geq 2\lambda C(C-1)$$

Since $b_i^\top M b_i = C$, we have $\|b_i\| \leq \sqrt{C/\sigma_{min}(M)}$. Therefore the norm of the gradient is at least $2\lambda C(C-1)/\|b_i\| \geq 2\sqrt{2}\lambda \sigma_{min}(M)$, this contradicts with the assumption. The norm of the rows must all be bounded. $\square$

*Proof of Lemma C.7.* We have that $b_i^\top M b_i \leq 2$ implies $\|b_i\|^2 \leq 2/\sigma_{min}(M)$. The norm of the whole matrix is bounded by $\|B\|_F \leq \sqrt{\sum_{i=1}^m \|b_i\|^2} \leq \sqrt{2m/\sigma_{min}(M)}$, so by triangle inequality we have

$$\|\nabla F_{a^\star, \mu, \lambda}(B)\| \leq \|\nabla \mathcal{F}_{a^\star, \mu, \lambda, \delta}(B)\| + \delta \|B\|_F \leq \varepsilon + \delta \sqrt{2m/\sigma_{min}(M)}.$$

$\square$

Finally we are ready to prove Theorem C.5.

*Proof of Theorem C.5.* We will separate $B$ into two components $B_\mathcal{S} = BP_\mathcal{S}$ and $B_\perp = BP_{\mathcal{S}^\perp}$.

We will first show that $B_\mathcal{S}$ is close to the desirable solution. To do that we will use Theorem C.4 [14]. By the choice of $\varepsilon, \delta$, we know from Lemma C.7 that $\|\nabla F_{a^\star, \mu, \lambda}(B_\mathcal{S})\| \leq 2\delta \sqrt{2m/\sigma_{min}(M)}$. Also, $\nabla^2 F_{a^\star, \mu, \lambda}(B_\mathcal{S}) \geq \nabla^2 \mathcal{F}_{a^\star, \mu, \lambda}(B) - \delta \geq -\tau_0$. Therefore we know $B_\mathcal{S}$ must be of the form

$$[B_\mathcal{S}^\dagger]^\top = PD\Gamma B^\star + E_1,$$

where $\|E_1\| < \varepsilon_0/2$. Also at the same time from the proof of Theorem C.4 we know $B_\mathcal{S} = (PDO + E')W^\top$ where $PDO + E'$ has singular values close to 1. Therefore $\sigma_{min}(B_\mathcal{S}) \geq \sigma_{min}(W)/2 = 1/2\sqrt{\|M\|}$.

By Lemma C.6 we know $\|B_\perp\|_F \leq \varepsilon/\delta$. We apply inverse matrix perturbation (Theorem C.9) again, using $B = B_\mathcal{S} + B_\perp$, therefore we know

$$B^\dagger = B_\mathcal{S}^\dagger + E_2,$$

where $\|E_2\| \leq O(\|B_\perp\|_F/\sigma_{min}^2(B_\mathcal{S})) \leq \varepsilon_0/2$.

Combining these two perturbations we know

$$[B^\dagger]^\top = PD\Gamma B^\star + E_1 + E_2^\top,$$

and the error term $E_1 + E_2^\top$ has spectral norm at most $\varepsilon_0$. $\square$

## C.4 TOOLBOX: MATRIX PERTURBATION

In the proof we used the following theorem for the perturbation of matrices.

**Theorem C.9** (Stewart and Sun Stewart & guang Sun (1990)). *Consider the perturbation of a matrix A: if $B = A + E$, then we have*

$$\|B^\dagger - A^\dagger\| \leq \sqrt{2}\|A^\dagger\|\|B^\dagger\|\|E\|.$$

*As a corollary, if $\|E\| \leq \sigma_{min}(A)/2$, then we have*

$$\|B^\dagger - A^\dagger\| \leq 2\sqrt{2}\sigma_{min}(A)^{-2}\|E\|.$$

---

[14]If we restrict all the vectors to the subspace $\mathcal{S}$, we can still apply Theorem C.4 as long as we replace all inverses with pseudo-inverses.

# D    RECOVERING THE LINEAR LAYER

We will show in this section that if we have are given a $\delta$-approximation of $B^\star$, then it is easy to recover $a^\star$. The key observation here is that the correlation between the $\langle b_i^\star, x \rangle$ and the output $y$ is exactly proportional to $a_i^\star$. We also note that there could be multiple other ways to recover $a^\star$, e.g., using linear regression with the $\sigma(Bx)$ as input and the $y$ as output. We chose this algorithm mostly because of the ease of analysis.

---

**Algorithm 1** Recovering $a^\star$

---

**Input:**  A matrix $B$ with unit row norms that is row-wise $\delta$-close to $B^\star$ in Euclidean distance.
**Return:**  Let $a_i' = 2\widehat{\mathbb{E}}[y\langle x, b_i \rangle]$ where $\widehat{\mathbb{E}}$ means the empirical average. Set $a_i \leftarrow |a_i'|$ and $b_i \leftarrow b_i \mathrm{sgn}(a_i')$

---

**Lemma** (Restatement of Lemma 2.5).  *Given a matrix $B$ whose rows are $\delta$-close to $B^\star$ in Euclidean distance up to permutation and sign flip with $\delta \leq 1/(2\kappa^\star)$. Then, we can give estimates $a, B'$ (using e.g., Algorithm 1) such that there exists a permutation $P$ where $\|a - Pa^\star\|_\infty \leq \delta a_{\max}^\star$ and $B'$ is row-wise $\delta$-close to $PB^\star$.*

To see why this simple algorithm works for recovering $a^\star$, we need the following simple claim.
*Claim* D.1.  For any vector $v$ we have

$$\mathbb{E}[y\langle x, v \rangle] = \frac{1}{2} \sum_{i=1}^m a_i^\star \langle b_i^\star, v \rangle.$$

The proof of this claim follows immediately from the property of Hermite polynomials. Now we are ready to prove the corollary.

*Proof.*  Without loss of generality we assume $B$ is close to a sign flip of $B^\star$. The unknown permutation does not change the proof.

Since $b_i$ is $\delta$ close to $B_i^\star$, let $u$ be the vector where $u_j = \langle b_j^\star, b_i - b_i^\star \rangle$, we have

$$a_i' = \sum_{i=1}^m a_i^\star \langle b_i^\star, b_i \rangle = \sum_{i=1}^m a_i^\star (\langle b_i^\star, b_i^\star \rangle + \langle b_i^\star, b_i - b_i^\star \rangle) = a_i^\star + \langle a_i^\star, u \rangle \in a_i^\star \pm a_{max}^\star \delta.$$

Therefore $a_i'$ is always positive, $a_i$ is in the desirable range and $\|B_i' - B_i^\star\| \leq \delta$.

Similarly, if $-b_i$ is $\delta$ close to $B_i^\star$, we have $a_i' \in -a_i^\star \pm a_{max}^\star \delta$, and the conclusion still holds.

$\square$

For the settings considered in Section C, the vectors $b_i^\star$ are not necessarily orthogonal. In this case we use the following algorithm:

---

**Algorithm 2** Recovering $a^\star$ for general case

---

**Input:**  A matrix $B$ with unit row norms, and $B$ is $\delta$-close to $B^\star$ in spectral norm up to permutation and sign flip.
Let $u_i = 2\widehat{\mathbb{E}}[y\langle x, b_i \rangle]$ where $\widehat{\mathbb{E}}$ means the empirical average.
Let $a' = (BB^\top)^{-1} u$.
**Return:** Set $a_i \leftarrow |a_i'|$ and $b_i \leftarrow b_i \mathrm{sgn}(a_i')$

---

**Lemma D.2.**  *Given a matrix $B$ whose rows have unit norm, and $\|B - SPB^\star\| \leq \delta$ for some permutation matrix $P$ and diagonal matrix $S$ with $\pm 1$ entries on diagonals. If $\frac{\sigma_{min}^2(B)}{4\sqrt{2}\kappa^\star \sqrt{m}}$, we can give estimates $a, B'$ (using e.g., Algorithm 2) such that $\|a - Pa^\star\| \leq \frac{2\sqrt{2}a_{max}^\star \sqrt{m}}{\sigma_{min}^{-2}(B)} \cdot \delta$ and $\|B' - PB^\star\| \leq \delta$.*

*Proof.* We again use Claim D.1: in this case we know the vector $u$ satisfies $u = B(B^\star)^\top a^\star$. As a result, for the vector $a'$, we have

$$a' = (BB^\top)^{-1}(B(B^\star)^\top)a^\star = (B^\dagger)^\top (B^\star)^\top a^\star = (B^\star B^\dagger)^\top a^\star.$$

By assumption we know $B = SPB^\star + E$ where $\|E\| \leq \delta$. By the perturbation of matrix inverse (Theorem C.9), we know if $\|E\| \leq \delta \leq \sigma_{min}(B)/2$, then $B^\dagger = (B^\star)^\dagger P^{-1}S^{-1} + E'$ where $\|E'\| \leq 2\sqrt{2}\sigma_{min}(B)^{-2}\delta$. Therefore

$$a' = (P^{-1}S^{-1} + E')^\top a^\star = S^{-\top}P^{-\top}a^\star + (E')^\top a^\star = SPa + (E')^\top a^\star.$$

(Here the last equality is because for both permutation matrix $P$ and sign flip matrix $S$, $P^{-\top} = P$ and $S^{-\top} = S$.) Therefore, coordinates of $a'$ are permutation and sign flips of $a^\star$, up to an error term $(E')^\top a^\star$.

When $\delta \leq \frac{\sigma_{min}^2(B)}{4\sqrt{2}\kappa^\star \sqrt{m}}$, we know $\|(E')^\top a^\star\| \leq \|E'\| a_{max}^\star \sqrt{m} \leq a_{min}^\star/2$, therefore the signs are all recovered correctly. After fixing the sign, we have $\|a - Pa\| \leq \|(E')^\top a^\star\| \leq \frac{2\sqrt{2}\delta a_{max}^\star \sqrt{m}}{\sigma_{min}^{-2}(B)}$, and $\|B' - PB^\star\| \leq \delta$. $\qquad\square$

# E    SAMPLE COMPLEXITY

In this section we will show that our algorithm only requires polynomially many samples to find the desired solution. Note that we did not try to optimize the polynomial dependency.

**Theorem E.1** (Theorem 2.7 Restated). *In the setting of Theorem 2.3, suppose we use $N$ empirical samples to approximate $G$ and obtain function $\widehat{G}$. There exists a fixed polynomial such that if $N \geq poly(d, a_{max}^\star/a_{min}^\star, 1/\varepsilon)$, with high probability for any point $B$ with $\lambda_{min}(\nabla^2\widehat{G}(B)) \geq -\tau_0/2$ and $\|\nabla\widehat{G}(B)\| \leq \varepsilon/2$, then $B$ can be written as $B = DP + E$ where $P$ is a permutation matrix, $D$ is a diagonal matrix and $|E|_\infty \leq O(\varepsilon/(\hat{\sigma}_4 a_{\min}^\star))$.*

In order to bound the sample complexity, we will prove a uniform convergence result: we show that with polynomially many samples, the gradient and Hessian of $\widehat{G}$ are point-wise close to the gradient and Hessian of $G$, therefore any approximate local minimum of $\widehat{G}$ must also be an approximate local minimum of $G$.

However, there are two technical issues in showing the uniform convergence result. The first issue is that when the norm of $B$ is very large, both the gradient and Hessian of $G$ and $\widehat{G}$ are very large and we cannot hope for good concentration. We deal with this issue by showing when $B$ has a large norm, the empirical gradient $\nabla\widehat{G}(B)$ must also have large norm, and therefore it can never be an approximate local minimum (we do this later in Lemma E.5). The second issue is that our objective function involves high-degree polynomials over Gaussian variables $x, y$, and is therefore not sub-Gaussian or sub-exponential. We use a standard truncation argument to show that the function does not change by too much if we restrict to the event that the Gaussian variables have bounded norm.

**Lemma E.2.** *Suppose $P'(B) + R(B) = \mathbb{E}_{(x,y)}[f(x, y, B)]$ where $f$ is a polynomial of degree at most $5$ in $x, y$ and at most $4$ in $B$. Also assume that the sum of absolute values of coefficients is bounded by $\Gamma$. For any $\varepsilon \leq \Gamma/2$, let $R = Cd\log(a_{max}^\star\Gamma/\varepsilon)$ for a large enough constant $C$, let $\mathcal{F}$ be the event that $\|x\|^2 \leq R$, and let $G_{trunc} = \mathbb{E}_{(x,y)}[f(x, y, B)1_\mathcal{F}]$. For any $B$ such that $\|b_i\| \leq 2$ for all rows, we have*

$$\|\nabla G(B) - \nabla G_{trunc}(B)\| \leq \varepsilon,$$

*and*

$$\|\nabla^2 G(B) - \nabla^2 G_{trunc}(B)\| \leq \varepsilon,$$

*Proof.* By standard $\chi^2$ concentration bounds, for large enough $C$ and any $z > R$, the probability that $\|x\|^2 \geq z$ is at most $\exp(-10z)$.

By simple calculation, it is easy to check that $\|\nabla_B f(x, y, B)\| \leq 4\Gamma d^{1.5}a_{max}^\star\|x\|^5$, and $\|\nabla_B^2 f(x, y, B)\| \leq 12\Gamma d^2 a_{max}^\star\|x\|^5$. We know $\|\nabla G(B) - \nabla G_{trunc}(B)\| = \|\mathbb{E}[\nabla_B f(x, y, B)(1-$

$1_{\mathcal{F}})\|$. The expectation between $\|x\|^2 \in [2^i R, 2^{i+1} R]$, for $i = 0, 1, 2, ...$, is always bounded by $4\Gamma d^{1.5} a_{max}^\star \|2^{i+1}R\|^5 \exp(-2^i R) < \varepsilon/2^{i+1}$. Therefore

$$\|\nabla G(B) - \nabla G_{trunc}(B)\| \leq \sum_{i=0}^{\infty} \varepsilon/2^{i+1} \leq \varepsilon.$$

The bound for the Hessian follows from the same argument. $\square$

Finally, we combine this truncation with a result of Mei et al. (2016) that proves universal convergence of gradient and Hessian. For completeness here we state a version of their theorem with bounded gradient/Hessian:

**Theorem E.3** (Theorem 1 in Mei et al. (2016)). *Let $f(\theta)$ be a function from $\mathbb{R}^p \to \mathbb{R}$ and $\hat{f}$ be its empirical version. If the norm of the gradient and Hessian of a function is always bounded by $\tau$, for variables in a ball of radius $r$ in $p$ dimensions, there exists a universal constant $C_0$ such that for $C = C_0 \max\{\log r\tau/\delta, 1\}$, the following hold:*

*(a) The sample gradient converges to the population gradient. Namely if $N \geq Cp\log p$ we have*

$$\Pr[\sup_{\|\theta\| \leq r} \|\nabla f(\theta) - \nabla \hat{f}_\theta\| \leq \tau\sqrt{\frac{Cp\log n}{n}}] \geq 1 - \delta.$$

*(b) The sample Hessian converges to the empirical Hessian. Namely if $N \geq Cp\log p$ we have*

$$\Pr[\sup_{\|\theta\| \leq r} \|\nabla f(\theta) - \nabla \hat{f}_\theta\| \leq \tau\sqrt{\frac{Cp\log n}{n}}] \geq 1 - \delta.$$

As an immediate corollary of this theorem and Lemma E.2, we have

**Corollary E.4.** *In the setting of Theorem 2.7, for every $B$ whose rows have norm at most 2, we have with high probability,*

$$\|\nabla G(B) - \nabla \widehat{G}(B)\| \leq \varepsilon/2,$$

*and*

$$\|\nabla^2 G(B) - \nabla^2 \widehat{G}(B)\| \leq \tau_0/2.$$

*Proof.* On the other hand, for all such matrices $B$, by Lemma E.2 we know the gradient and Hessian of $G$ is close to the gradient and Hessian of $G_{trunc}$.

$$\|\nabla G(B) - \nabla G_{trunc}(B)\| \leq \varepsilon/4,$$

and

$$\|\nabla^2 G(B) - \nabla^2 G_{trunc}(B)\| \leq \tau_0/4.$$

Now, the gradient and Hessian for individual samples for estimating $G_{trunc}$ are bounded by some $\text{poly}(d, 1/\varepsilon)$, therefore by Theorem E.3 we know the gradient and Hessian of $\widehat{G}$ are close to those of $G_{trunc}$. When $N \geq \text{poly}(d, 1/\varepsilon)$ for a large enough polynomial, we have with high probability, for all $B$ with all rows $\|b_i\| \leq 2$,

$$\|\nabla G_{trunc}(B) - \nabla \widehat{G}(B)\| \leq \varepsilon/4,$$

and

$$\|\nabla^2 G_{trunc}(B) - \nabla^2 \widehat{G}(B)\| \leq \tau_0/4.$$

The corollary then follows from triangle inequality. $\square$

Finally we handle the case when $B$ has a row with large norm. We will show that in this case $\nabla \widehat{G}(B)$ must also be large, so $B$ cannot be an approximate local minimum.

**Lemma E.5.** *If $b_i$ is the row with largest norm and $\|b_i\| \geq 2$, then when $N \geq poly(d, a_{max}^\star/a_{min}^\star)$ for some fixed polynomial, we have with high probability $\langle \nabla \widehat{G}(B), b_i \rangle \geq c\lambda \|b_i\|^4$ for some universal constant $c > 0$.*

*Proof.* The proof of this Lemma is very similar to Claim C.8. Note that by equation (2.7) there are three terms in $\widehat{G}(B)$:(1) $\mathrm{sign}(\hat{\sigma}_4)\hat{\mathbb{E}}\left[y \cdot \sum_{j,k \in [d], j \neq k} \phi(b_j, b_k, x)\right]$, (2) $-\mu \, \mathrm{sign}(\hat{\sigma}_4)\hat{\mathbb{E}}\left[y \cdot \sum_{j \in [d]} \varphi(b_j, x)\right]$, (3) $\lambda \sum_{i=1}^m (\|b_i\|^2 - 1)^2$. Here $\hat{E}$ is the empirical average over the samples.

Note that the first two terms are homogeneous degree 4 polynomials over $B$, and the third term does not depend on the sample. By argument similar to Corollary E.4, we know for any $B$ where $b_i$ has the largest row norm, with the number of samples we choose the gradient of the first two terms is $c a^\star_{min}\|b_i\|^3$ close to the gradient of their expectations, where $c < 0.01$ is a small constant.

By Theorem 2.6, we know the expectation of the first two terms are equal to $A1(B) = 2\sqrt{6}|\hat{\sigma}_4| \cdot \sum_{i \in [d]} a^\star_i \sum_{j,k \in [d], j \neq k} \langle b^\star_i, b_j \rangle^2 \langle b^\star_i, b_k \rangle^2$ and $A2(B) = -\frac{|\hat{\sigma}_4|\mu}{\sqrt{6}} \sum_{i,j \in [d]} a^\star_i \langle b^\star_i, b_j \rangle^4$. Here the gradient of the first term always have positive correlation with $b_i$, so we can ignore it. For the second term, we know the gradient

$$\frac{\partial}{\partial b_i}[A2(B)] = -\frac{|\hat{\sigma}_4|\mu}{\sqrt{6}} \sum_j a^\star_j \langle b^\star_j, b_i \rangle^3 b^\star_j.$$

Taking the inner-product with $b_i$, and use the fact that $b^\star_i$ form an orthonormal basis, we know

$$\langle \frac{\partial}{\partial b_i}[A2(B)], b_i \rangle \geq -c a^\star_{min}\|b_i\|^4.$$

On the other hand, when $\|b_i\| \geq 2$, we have for the third term

$$\langle \frac{\partial}{\partial b_i}[\lambda(\|b_i\|^2 - 1)^2], b_i \rangle \geq \lambda(\|b_i\| - 1)^4 \geq \lambda\|b_i\|^4/16.$$

Since $\lambda$ is larger than $a^\star_{max}$, we know the negative contribution from $A2$ and the difference between the empirical version and $G$ are both negligible. Therefore we have $\langle \nabla\widehat{G}(B), b_i \rangle \geq c\lambda\|b_i\|^4$ as desired. □

Now we are ready to prove Theorem 2.7:

*Proof.* By Lemma E.5, any point $B$ with $\nabla\widehat{G}(B) \leq \varepsilon$ must have $\|b_i\| \leq 2$ for all $i$. Now by Corollary E.4, we know the point $B$ we have must satisfy

$$\|\nabla G(B)\| \leq \varepsilon; \nabla^2 G(B) \succeq -\tau_0 \mathrm{Id}.$$

By point 3 in Theorem 2.3, this implies the guarantee on $B$. □

# F SPURIOUS LOCAL MINIMUM FOR FUNCTION $P'$

In this section we give an example where the function $P'$ does have spurious local minimum.

In this example, $d = 4$, and the true vectors are the standard basis vectors $b^\star_i = e_i$. We will set $a^\star_1 = 1$, and $a^\star_2 = a^\star_3 = a^\star_4 = 2 + \delta$ (where $\delta > 0$ is an arbitrary positive constant).

The spurious local minimum that we consider is $b_1 = b_2 = e_1 = b^\star_1$, $b_3 = e_2 = b^\star_2$, $b_4 = \frac{\sqrt{2}}{2}e_3 + \frac{\sqrt{2}}{2}e_4$. That is,

$$B = \begin{pmatrix} 1 & 0 & 0 & 0 \\ 1 & 0 & 0 & 0 \\ 0 & 1 & 0 & 0 \\ 0 & 0 & \frac{\sqrt{2}}{2} & \frac{\sqrt{2}}{2} \end{pmatrix}.$$

The objective $P'(B) = 1$ and the only non-zero term is $a^\star_1 \langle b^\star_1, b_1 \rangle^2 \langle b^\star_1, b_2 \rangle^2$. In order to improve the objective locally, we need to change either $b_1$ or $b_2$, otherwise the term $a^\star_1 \langle b^\star_1, b_1 \rangle^2 \langle b^\star_1, b_2 \rangle^2$ is still 1, and all other terms ($a^\star_i \langle b^\star_i, b_j \rangle^2 \langle b^\star_i, b_k \rangle^2$) are non-negative.

Assume we have a local perturbation $B'$, where $b'_1 = \sqrt{1 - \varepsilon_1^2} e_1 + \varepsilon_1 u_1$, $b'_2 = \sqrt{1 - \varepsilon_2^2} e_1 + \varepsilon_2 u_2$. Here $u_1, u_2$ are unit vectors that are orthogonal to $e_1$. Also, since this is a local perturbation, we make sure $\varepsilon_1, \varepsilon_2 \leq \varepsilon$, and $b_3(2) \geq 1 - \varepsilon$, $[b_4(3)]^2, [b_4(4)]^2 \geq 0.5 - \varepsilon$. We will show that when $\varepsilon$ is small enough, the objective function $P'(B') \geq 1$.

To see this, notice that the term $a_1^\star \langle b_1^\star, b_1 \rangle^2 \langle b_1^\star, b_2 \rangle^2$ is now equal to $(1 - \varepsilon_1^2)(1 - \varepsilon_2^2)$. On the other hand, for $b_1$, we have

$$
\begin{aligned}
\sum_{i=2}^{4} a_i^\star \sum_{k=3}^{4} \langle b_i^\star, b_1 \rangle^2 \langle b_i^\star, b_k \rangle^2 &= \varepsilon_1^2 (2 + \delta) \sum_{i=2}^{4} \sum_{k=3}^{4} \langle b_i^\star, u_1 \rangle^2 \langle b_i^\star, b_k \rangle^2 \\
&= \varepsilon_1^2 (2 + \delta) \sum_{i=2}^{4} \langle b_i^\star, u_1 \rangle^2 \left( \sum_{k=3}^{4} \langle b_i^\star, b_k \rangle^2 \right) \\
&\geq \varepsilon_1^2 (2 + \delta) \sum_{i=2}^{4} \langle b_i^\star, u_1 \rangle^2 \cdot \min_{i=2}^{4} \left\{ \sum_{k=3}^{4} \langle b_i^\star, b_k \rangle^2 \right\} \\
&\geq \varepsilon_1^2 (2 + \delta)(0.5 - \varepsilon).
\end{aligned}
$$

Similarly we have the same equation for $b_2$. Note that all the terms we analyzed are disjoint, therefore

$$
P'(B') \geq (1 - \varepsilon_1^2)(1 - \varepsilon_2^2) + \varepsilon_1^2 (2 + \delta)(0.5 - \varepsilon) + \varepsilon_2^2 (2 + \delta)(0.5 - \varepsilon).
$$

By removing higher order terms of $\varepsilon$, it is easy to see that $P'(B') \geq 1$ when $\varepsilon$ is small enough. Therefore $B$ is a local minima of $P'$.

