# OpenReview forum: "Learning One-hidden-layer Neural Networks with Landscape Design"
_ICLR.cc/2018/Conference — Accept (Poster)_

### Official Review · AnonReviewer2 · 2017-11-26
**Overall an interesting heuristic approach to solving SGD problems on one particular type of data, with limited practical applications**

**Rating:** 6
**Confidence:** 3

**Review:**

[ =========================== REVISION ===============================================================]
I am satisfied with the answers to my questions. The paper still needs some work on clarity, and authors defer the changes to the next version (but as I understood, they did no changes for this paper as of now), which is a bit frustrating. However I am fine accepting it.
[ ============================== END OF REVISION =====================================================]

This paper concerns with addressing the issue of SGD not converging to the optimal parameters on one hidden layer network for a particular type of data and label (gaussian features, label generated using a particular function that should be learnable with neural net). Authors demonstrate empirically that this particular learning problem is hard for SGD with l2 loss (due to apparently bad local optima) and suggest two ways of addressing it, on top of the known way of dealing with this problem (which is overparameterization). First is to use a new activation function, the second is by designing a new objective function that has only global optima and which can be efficiently learnt with SGD

Overall the paper is well written. The authors first introduce their suggested loss function and then go into details about what inspired its creation. I do find interesting the formulation of population risk in terms of tensor decomposition, this is insightful

My issues with the paper are as follows:
- The loss function designed seems overly complicated. On top of that authors notice that to learn with this loss efficiently, much larger batches had to be used. I wonder how applicable this in practice - I frankly didn't see insights here that I can apply to other problems that don't fit into this particular narrowly defined framework
- I do find it somewhat strange that no insight to the actual problem is provided (e.g. it is known empirically but there is no explanation of what actually happens and there is a idea that it is due to local optima), but authors are concerned with developing new loss function that has provable properties about global optima. Since it is all empirical, the first fix (activation function) seems sufficient to me and new loss is very far-fetched.
- It seems that changing activation function from relu to their proposed one fixes the problem without their new loss, so i wonder whether it is a problem with relu itself and may be other activations funcs, like sigmoids will not suffer from the same problem
- No comparison with overparameterization in experiments results is given, which makes me wonder why their method is better.

Minor: fix margins in formula 2.7.

---

> ### Author Response · Authors · 2017-12-21
> **response**
>
> Thanks for the review and comments.
>
> Response to your questions:
>
> --- "empirical fix seems sufficient": We do consider the proposal of the empirical fix as part of the contribution of the paper. It's driven by the theoretical analysis of the squared loss (more intuition below), and it's novel as far as we know.
>
> That said, we think it's also valuable to pursue a provably nice loss function without bad local minima. Note that there is no known method to empirically verify whether a function has no bad local minima, and such statements can only be established by proofs. Admittedly our first-cut design of the loss function is complicated and sub-optimal in terms of sample complexity, we do hope that our technique can inspire better loss function and model design in the future.
>
> --- "does sigmoid suffer from the same problem?": we did experiment with the sigmoid activation and found that the sigmoid activation function also has bad local minima. We will add this experiment to the next version of the paper. We conjecture that the activation h_2 +h_4 or the activation 1/2 |z|  has no spurious local minima.
>
> --- "actual insights about the landscape": Our insights is that the squared loss objective function is trying to perform an infinite number of tensor decomposition problems simultaneously, and the mixing of all of these problems very likely creates bad local minima, as we empirically observed. The intuition behind the empirical fix is that removing some of these tensor decomposition problems would make the landscape simpler and nicer.
>
> --- "comparison with over-parameterization":  over-parameterization is indeed a powerful way to remove the bad local minima, and it gives models with good prediction. But it doesn't recover the parameters of the true model because the training parameters space is larger. Our method is guaranteed to recover the true parameters of the model, which in turns guarantees a ``complete" generalization to any unseen examples, even including e.g., adversarial examples or test examples drawn from another distribution. In this sense, our approaches (both the empirical fix and theoretical one) return solutions with stronger guarantees than over-parameterization can provide.
>
> Of course, it's also a very important open problem to understand better other alternatives to landscape design such as over-parametrization.

---

> ### Author Response · Authors · 2018-01-12
> **revision**
>
> Thanks for the review again!
>
> We apologize that we didn't know that the paper was expected to be updated. We just added the results for sigmoid that answers the question "does sigmoid suffer from the same problem?" as we claimed in the response before. Please see page 9, figure 2 in the current version.
>
> We will revise the paper with more intuitions/explanations as promised in the previous response as soon as possible.

---

### Official Review · AnonReviewer1 · 2017-11-27
**Interesting paper and solid contribution: Accept.**

**Rating:** 9
**Confidence:** 3

**Review:**

This paper studies the problem of learning one-hidden layer neural networks and is a theory paper. A well-known problem is that without good initialization, it is not easy to learn the hidden parameters via gradient descent. This paper establishes an interesting connection between least squares population loss and Hermite polynomials. Following from this connection authors propose a new loss function. Interestingly, they are able to show that the loss function globally converges to the hidden weight matrix. Simulations confirm the findings.

Overall, pretty interesting result and solid contribution. The paper also raises good questions for future works. For instance, is designing alternative loss function useful in practice? In summary, I recommend acceptance. The paper seems rushed to me so authors should polish up the paper and fix typos.

Two questions:
1) Authors do not require a^* to recover B^*. Is that because B^* is assumed to have unit length rows? If so they should clarify this otherwise it confuses the reader a bit.
2) What can be said about rate of convergence in terms of network parameters? Currently a generic bound is employed which is not very insightful in my opinion.

---

> ### Author Response · Authors · 2017-12-21
> **response**
>
> Thanks for the comments.
> Regarding the questions:
>
> 1) Yes, since we mostly focus on the ReLU activation, we assume that the rows of B^* have unit norms. For ReLU activation, the row norms are inherently unidentifiable.
>
> 2) The technical version of the landscape analysis in Theorem B.1 specifies the precise dependencies of the landscape properties on the dimension, etc. To get a convergence rate, one can combine our Theorem B.1 with an analysis of gradient descent or other algorithms on non-convex functions. The best-known analysis for SGD in Ge et al. 2015 does not specify the precise polynomial dependencies. Since developing stochastic algorithms (beyond SGD) with lower iteration complexity is an active area of research, the best-known convergence rate is constantly changing.

---

### Official Review · AnonReviewer3 · 2017-11-27
**An interesting tensor factorization-type method for learning one hidden-layer neural network**

**Rating:** 7
**Confidence:** 3

**Review:**

This paper proposes a tensor factorization-type method for learning one hidden-layer neural network. The most interesting part is the Hermite polynomial expansion of the activation function. Such a decomposition allows them to convert the population risk function as a fourth-order orthogonal tensor factorization problem. They further redesign a new formulation for the tensor decomposition problem, and show that the new formulation enjoys the nice strict saddle properties as shown in Ge et al. 2015. At last, they also establish the sample complexity for recovery.

The organization and presentation of the paper need some improvement. For example, the authors defer many technical details. To make the paper accessible to the readers, they could provide more intuitions in the first 9 pages.

There are also some typos: For example, the dimension of a is inconsistent. In the abstract, a is an m-dimensional vector, and on Page 2, a is a d-dimensional vector. On Page 8, P(B) should be a degree-4 polynomial of B.

The paper does not contains any experimental results on real data.

---

> ### Author Response · Authors · 2017-12-21
> **response**
>
> Thanks for the comments. We will add more intuitions in the paper and fix the typos.
>
> The high-level intuition is that the squared loss objective function is trying to perform an infinite number of tensor decomposition problems simultaneously and the mixing of all of these problems very likely creates bad local minima, as we empirically observed. Thus we design an objective function that selects only two of these tensor decompositions problems, which empirically removes the bad local minima. Finally, we design an objective function that resembles more a 4th order tensor decomposition objective in Ge et al. (2015), which are known to be good.

---

### Public Comment · (anonymous) · 2018-02-19
**Some inconsistencies in the notations and inclusion of bias parameter**

I really liked reading the paper and the ideas behind the design of a new loss function to eliminate the bad landscape properties inherent to $l_2$-loss function. This idea might be useful in other domains too. However, I have some doubts about the paper.

1) I think there are still notation inconsistencies in the revised paper and it's hard to follow. For example, till Section 2.2 it was assumed that the matrix is of size $m*d$ where $m \leq d$. However, from Section 2.2 on wards when it is assumed that $B^\ast$ is an orthogonal matrix, is it with $m=d$ or with $m <d$ and rows being orthogonal to each other? This confusion also arises in the statement of Theorem 2.6 where the notation indices in the first term of $G(B)$ are all in terms of $d$ whereas the last term has indices in $m$ though they both correspond to the same variable $B$. So I am wondering if the said results in this paper only hold for the case $m=d$ and $B^\ast$ is orthogonal or they continue to hold even if $m <d$ and the rows are orthogonal to each other. I think it would be better if authors can follow a consistent notation through out the paper.

2) In typical neural networks, in addition to the weight matrices, there is always a bias term present which is conspicuous by absence in this paper. So I am wondering if one introduces a bias vector $b in \mathbb{R}^m$ in equation (1.1), do the results still hold? I suspect that inclusion of a bias term might make the analysis harder since the Hermite polynomial trick might no longer be valid. I would like to know author's thoughts on this and if this issue can be dealt with some other way.

---

### Decision · Program_Chairs · 2018-01-29
**ICLR 2018 Conference Acceptance Decision**

**Decision:**

Accept (Poster)

**Comment:**

I recommend acceptance based on the reviews. The paper makes novel contributions to learning one-hidden layer neural networks and designing new objective function with no bad local optima.

 There is one point that the paper is missing. It only mentions Janzamin et al in the passing. Janzamin et al propose using score function framework for designing alternative objective function. For the case of Gaussian input that this paper considers, the score function reduces to Hermite polynomials. Lack of discussion about this connection is weird. There should be proper acknowledgement of prior work. Also missing are some of the key papers on tensor decomposition and its analysis

I think there are enough contributions in the paper for acceptance irrespective of the above aspect.